🔓 | **Open Peer Review** | Computational Biology | Research Article

# Probing the genomic and proteomic basis of encystment in *Oxytricha granulifera*

Miaomiao Wang,[1] Juan Yang,[2] Tao Hu,[1] Zina Lin,[1] Tian Wang,[1] ZiJia Liu,[3] Xiao Chen,[2] Xinpeng Fan[1]

**ABSTRACT** Protozoan encystment constitutes a pivotal survival strategy against environmental stressors; however, the molecular architecture governing this transition remains enigmatic, owing to limited genomic resources and a scarcity of integrated multi-omics investigations. Here, we elucidate the mechanisms underlying encystment in *Oxytricha granulifera* by reporting the first macronuclear genome assembly and conducting a comprehensive integration of transcriptomic, proteomic, and morphological analyses across vegetative and cyst stages. Morphological restructuring, typified by ciliary dedifferentiation and cyst wall formation, is molecularly supported by the downregulation of microtubule dynamics-associated genes and the concurrent upregulation of vesicle transport machinery. Furthermore, expanded gene families linked to carbohydrate metabolism and cellular acidification coincide with observed autophagic clearance and mucocyst activity, highlighting a coordinated metabolic shift essential for cyst formation. Elevated expression of the ubiquitin-proteasome system and autophagy pathways, which mediate protein turnover, along with upregulation of antioxidant enzyme genes, contributes to alleviating oxidative damage. Notably, we identified rewired post-transcriptional regulation that increases spliceosome activity and alternative splicing frequency, with each trend validated at the protein level. Concurrently, we observed a distinct epigenetic signature characterized by the significant downregulation of DNA $N^6$-adenine methylation (6mA) methyltransferases (homologs of AMT1 and AMT6/7), suggesting a potential repressive role of methylation during the cyst stage. Collectively, these findings provide a multidimensional atlas of the encystment process, revealing that *O. granulifera* accomplishes cellular structural remodeling through a multilayered regulatory network spanning morphological, genetic, transcriptomic, and proteomic levels.

**IMPORTANCE** *Oxytricha* species are widely distributed in freshwater and terrestrial ecosystems, playing significant ecological roles in microbial communities. Their ability to undergo encystment provides a powerful model for studying cellular differentiation and stress adaptation in microbial eukaryotes. This study presents the first multi-omics analysis of encystment in *Oxytricha granulifera*, revealing microbial survival strategies through enhanced protein turnover, autophagy, alternative splicing, and DNA methylation reprogramming. These findings offer fundamental insights into dormancy mechanisms and environmental adaptation in protists, advancing our understanding of microbial resilience, evolutionary innovation, and ecological success in fluctuating environments.

**KEYWORDS** cyst, mucocyst, spliceosome, transcriptomic, 6mA

D ormancy is a crucial survival strategy for protozoans, enabling them to endure harsh conditions through a state of reduced metabolic activity. Widespread among protists, this phenomenon forms a "seed bank," which enables them to disperse to new

**Peer Reviewer** Vojtech Zarsky, University of Ostrava, Ostrava, Czechia

Address correspondence to Xiao Chen, xc@sdu.edu.cn, or Xinpeng Fan, xpfan@bio.ecnu.edu.cn.

Miaomiao Wang and Juan Yang contributed equally to this article. Author order was determined by drawing straws.

The authors declare no conflict of interest.

See the funding table on p. 20.

10.1128/msystems.01757-25   **1**

locations and contribute to genetic variation when they resume growth (1, 2). It also impacts interactions between parasitic protozoans and their hosts, such as the ability of these pathogens to persist within hosts and resist treatments (3, 4). Protozoan dormancy is characterized by reduced metabolic activity, increased stress tolerance, cessation of growth and division, and sometimes the formation of specialized structures (5, 6). These characteristics enable protozoans to survive unfavorable conditions and preserve genetic diversity. The molecular mechanisms underlying protozoan dormancy involve several key processes. Regulatory proteins, such as hibernation factors, bind to essential cellular components like ribosomes and RNA polymerase, preventing their degradation and maintaining a state of metabolic stasis (7). Additionally, dormancy can be triggered by the accumulation of specific signaling molecules that respond to environmental stressors (8). These mechanisms collectively allow protozoans to enter and maintain a dormant state until conditions become favorable for growth and activity. Understanding the dormancy mechanisms of protists is significant for comprehending their survival strategies, evolutionary adaptability, and ecological roles (9).

Ciliates, a globally distributed group of eukaryotic microbes, have evolved an extraordinary dormancy mechanism by forming cysts to cope with diverse environmental challenges (6). In recent decades, research into encystment of ciliates has focused on their drastic morphological and physiological changes, as well as underlying molecular mechanisms. Morphologically, encystment in ciliates involves several distinct stages. It typically begins with a reduction in cellular volume, followed by progressive sphericalization. The cyst wall, composed of distinct layers derived from different precursors, forms subsequently. During this process, ciliates exhibit varying degrees of ciliature resorption. Some lose both cilia and their supporting membranous structures, while others retain kinetosomes and/or microtubular structures (10, 11). During encystment, ciliates accumulate reserve grains and aggregate their mitochondria in the peripheral cytoplasmic region. The macronuclei fuse, and their chromatin condenses into small, spherical, dense bodies that are regularly scattered throughout the nuclear matrix. Additionally, various organelles within the cell undergo autophagy (3, 12).

These biological events are thought to be regulated by intracellular signaling pathways that transduce environmental cues into gene expression alterations. A previous study has suggested that *Colpoda* encystment may be mediated by elevated cAMP levels resulting from adenylate cyclase activation, which is potentially regulated by the $Ca^{2+}$/calmodulin complex (13). Comparative transcriptomic analyses between the vegetative and encystment stages are increasingly being utilized to elucidate the molecular regulatory mechanisms underlying these distinct physiological phases in a growing number of ciliate species, such as *Euplotes encysticus*, *Colpoda aspera*, *Pseudourostyla cristata*, and *Apodileptus* cf. *visscheri*. These studies have revealed significant pathway differences in cell cycle regulation, biosynthesis, and energy metabolism, while identifying key differentially expressed genes and signaling pathways—including cyclic adenosine monophosphate (cAMP), mammalian target of rapamycin (mTOR), phosphatidylinositol 3-kinase/protein kinase B (PI3K/AKT), calcium, and AMP-activated protein kinase (AMPK)—that potentially constitute a comprehensive regulatory network for encystment (14–19). Furthermore, Liu et al. (20) proposed a 6mA-mediated epigenetic regulatory mechanism in which dynamic methylation-demethylation cycling modulates gene expression to coordinate encystment-excystment transitions in ciliates, enabling rapid environmental adaptation with minimal energy cost.

Despite significant advances in understanding ciliate encystment at the morphological and transcriptomic levels, critical gaps remain in our comprehensive understanding of this crucial survival mechanism. Currently, three major limitations hinder progress in this field: (i) the overwhelming majority of studies have focused solely on transcriptomic changes, creating a significant protein-level validation gap for putative dormancy regulators. (ii) Genomic resources for encystment-capable ciliates remain remarkably scarce, with only a handful of species (e.g., *Pseudourostyla cristata*, *Colpoda steinii*) having available genome assemblies (21, 22). This severe limitation prevents meaningful

comparative genomic analyses to identify conserved encystment-related gene families and lineage-specific adaptations. (iii) Existing studies lack integration across biological scales—while morphological changes are well-documented and some molecular pathways have been identified, the crucial links between genomic features, protein networks, and cellular restructuring remain largely unexplored.

*Oxytricha* species are widely distributed in freshwater and terrestrial ecosystems, where they thrive in a variety of ecological niches. The ability of *Oxytricha* to undergo encystment makes it an excellent model for studying cellular differentiation, gene regulation, and survival strategies in eukaryotic microbes (23). Investigating the genetic and proteomic responses underlying encystment not only reveals how microorganisms adapt to environmental stressors but also provides broader insights into eukaryotic resilience and evolutionary innovation.

Here, the present study aims to (i) uncover the molecular basis through *de novo* macronuclear assembly and gene annotation of *O. granulifera* for the first time; (ii) systematically characterize the dynamic gene expression and protein profiles during the transition from vegetative growth to encystment by transcriptomic and proteomic analyses; (iii) identify the key molecular pathways involved in cyst formation through functional annotation of differentially expressed transcripts and proteins; and (iv) elucidate the key mechanisms of encystment and advance the understanding of stress adaptation in microbial eukaryotes by reconciling molecular processes with cellular morphological restructuring.

## MATERIALS AND METHODS

### Cell culture, encystment induction, and cell collection

*Oxytricha granulifera* was isolated from the soil samples collected in October 2019 from Tianmu Mountain, Zhejiang Province, China (30°19′45″N, 119°26′32″E). The vegetative cells were cultured in 500 mL sterile cell culture bottles containing Nongfu Spring mineral water, with *Chlamydomonas reinhardtii* as the sole food source, and a large number of vegetative cells (2,000 cells/mL) can be obtained in 3–7 days at room temperature. For genome, transcriptome, and proteome sequencing, vegetative cells were starved for 1 day and purified with nylon sieves with a pore size of 1 µm (cells passed through while impurities were trapped), followed by centrifugation at $70 \times g$ for 5 min at 18°C and removal of the supernatant. To induce the formation of mature cysts, the purified vegetative cells were transferred to Petri dishes and subjected to starvation conditions in culturing water for 30 days. Microscopic examination revealed a 100% encystment efficiency, demonstrating homogeneous conversion to mature cyst morphology. To confirm dormancy and viability, an excystment assay was performed, yielding an 85% success rate (17–18 out of 20 cysts), indicating that the cysts used for subsequent analyses were biologically active and in a true dormant state. The cysts were centrifuged at $1,377 \times g$ for 5 min at 4°C, after which the supernatant was carefully removed. For transcriptome and proteome sequencing, the purified vegetative cell and cyst samples were divided into the Cyst group (cysts) and the Veg group (vegetative cells), with three biological replicates per group, each containing approximately $1.5 \times 10^{6}$ cells.

### Morphological observation by light microscope and electron microscope

Staining of encysting cells with protargol was performed to reveal changes in the ciliature during encystment (24). Differential interference contrast (DIC) microscopy was performed using an Olympus BX 53 microscope equipped with a digital camera (Olympus DP 74).

For fluorescence microscopy, cells were fixed with Bouin's fixative at room temperature and washed with ultrapure water in an embryo dish. Then, they were stained by FITC-labeled concanavalin A (FITC-Con A) (Alpha Diagnostic International, San Antonio,

TX, USA) for 10 min and with DAPI (Beyotime, China) for 1 min, respectively. The cells were washed again before being transferred onto a slide. Cells were observed using a fluorescence microscope (Nikon Ni-U, Japan) equipped with fluorescent filter blocks for ex/em at 490/525 nm and 358/461 nm.

For transmission electron microscopy (25), 10-day-old encysting cells were fixed for 10 min at 4°C in a fixative of 2% $OsO_4$ and 6.25% glutaraldehyde, then washed with 0.2 M phosphate buffer and post-fixed in a fixative of 2% $O_SO_4$ and 3% potassium hexacyanoferrate for 30 min. The samples were washed again with phosphate buffer, dehydrated through a graded acetone series, and embedded in Epon 12 resin (Ted Pella). After ultrathin sectioning, samples were observed with a Hitachi JEM2100 transmission electron microscope at an accelerating voltage of 120 kV.

For scanning electron microscopy (26), samples were treated mainly according to the method described by Fan et al. (27). Cells were fixed in Párducz's fixative (1:4 mixture of 1% $O_SO_4$ and a saturated solution of $HgCl_2$) at room temperature for 10 min. The cells were then rinsed with 0.1 M phosphate buffer, dehydrated in a graded series of ethanol, dried with a critical point dryer (Leica CPD300), and coated with gold in an ion coater (Leica ACE600). Observations were performed using a Hitachi S-4800 at an accelerating voltage of 3 kV.

## Genomic DNA sequencing and *de novo* assembly

Genomic DNA was extracted from approximately $1 \times 10^6$ *O. granulifera* cells which were isolated from clonal cultures, using a sodium dodecyl sulfate (SDS)-based lysis method. The quality of the DNA was checked using a 1% agarose gel, and its purity was determined with a NanoDrop One UV-Vis spectrophotometer (Thermo Fisher Scientific, USA). The concentration of DNA was further measured using a Qubit Fluorometer (Invitrogen, USA). The extracted DNA was fragmented randomly into 350 bp fragments using Covaris ultrasonic crusher, and the ends were repaired. An adenine nucleotide was then added at the ends, and full-length adaptor sequences were connected. These libraries were purified using the AMPure XP system (Beckman Coulter, Brea, CA, USA). An Agilent 2100 Bioanalyzer and a real-time PCR were used for size distribution and quantitative analyses of the purified products. After the quality of the library was verified, all of the samples were subjected to paired-end sequencing by using the Illumina HiSeq 4000 platform with a read length of 150 base pairs (PE150). A total of 15 Gb of raw data were generated. Given the assembled genome size of 41.7 Mb, this corresponds to an estimated sequencing depth of approximately 360× for the macronuclear genome.

After low-quality reads were filtered using fastp v0.24.1 (default parameters) (19), the clean reads in the sequencing data were further evaluated with FastQC v0.11.9 (https://www.bioinformatics.babraham.ac.uk/projects/fastqc). The genome was assembled using SPAdes v3.14.0 (-k 21,33,55,77) (28). Contigs that were potential bacterial, *Chlamydomonas,* and mitochondrial contamination were removed by homologous search using BLAST v2.9.0+ (29). Due to the low GC content typically observed in ciliates, we removed sequences from the genome with a GC content higher than 50%, as they were considered potential contaminants based on previous research (30). In order to eliminate redundancy in the data set, we employed CD-HIT v.4.8.1 (31) with a sequence identity threshold set at 98%. Contigs that lacked sufficient support (coverage <5 or length <400 bp) were subsequently excluded.

Repetitive elements were identified using RepeatModeler2 (v2.0.3) for *de novo* repeat family discovery and RepeatMasker (v4.1.2) for genome-wide masking (32, 33). We used QUAST v5.0.1 (34) with default settings to obtain statistical information about the assembled genomes. MEME (35) was used for motif searching in the 5′ and 3′ subtelomeric regions. To assist gene structure prediction, transcriptome data (see Total RNA extraction, library preparation and assembly section) was assembled using Trinity v2.1 in genome-guided mode (parameters: --genome_guided_bam --jaccard_clip --genome_guided_max_intron 100 --full_cleanup) (36). Introns were identified using custom Perl scripts. Protein-coding gene models, including CDS and

transcript boundaries, were predicted by EuGene v1.5 (37), integrating evidence from the genome assembly and transcriptomic data. The completeness of the gene set was assessed using BUSCO v5.4.0 (38) against the Alveolata lineage data set. Functional annotation of protein-coding genes was performed by querying six databases: the Non-Redundant (8) database, SwissProt, eggNOG, InterPro, Gene Ontology (GO), and the Kyoto Encyclopedia of Genes and Genomes (KEGG).

## Total RNA extraction, library preparation

Total RNA was extracted using the mirVana miRNA Isolation Kit (Ambion-1561, Thermo Fisher Scientific, Waltham, MA, USA), following the manufacturer's instructions. Sequencing libraries were constructed using the TruSeq Stranded mRNA LT Sample Prep Kit (Illumina, San Diego, CA, USA) following the manufacturer's recommendations. Briefly, 4 µg total RNA was subjected to mRNA purification and fragmentation with RNA purification beads, followed by first-strand cDNA synthesis using SuperScript II Reverse Transcriptase (Invitrogen, 18064014) and second-strand cDNA synthesis with second strand marking master mix. The double-stranded cDNA was purified using Agencourt AMPure XP beads (Beckman Coulter, A63881) and then adenylated at the 3′ ends. RNA adapter indices were ligated to the cDNA fragments, and the ligation products were purified. Finally, the library fragments were amplified by PCR (15 cycles) and purified again. The libraries were sequenced on an Illumina platform. RNA integrity and concentration were assessed using an Agilent 2100 Bioanalyzer and NanoDrop 2000. On average, approximately 6 Gb of clean data (Q30 > 90%) was obtained for each sample.

## Differentially expressed gene analysis

Hisat2 (v2.1.1) was used to map clean reads to the assembled *O. granulifera* reference genome using the same parameters mentioned above (39). The mapped reads for each gene loci were counted using featureCounts (v1.3.1) (40). The R package "DESeq2" (41) was utilized to standardize and compare the gene expression levels between vegetative cells and dormant cysts of *O. granulifera*. The DESeq2 model internally applies the median-of-ratios method to normalize read counts for differences in library size (sequencing depth) across samples. This method does not rely on a specific reference sample but constructs a pseudo-reference from the geometric mean of all samples. Genes with an absolute $log_2$ fold change ≥1 (equivalent to a fold change of 2) and an adjusted *P*-value (Benjamini-Hochberg procedure) < 0.1 were defined as differentially expressed genes (DEGs). Principal component analysis (PCA) was carried out using the princomp function of the R package "ggplot2" v3.4.1 (42). Heat maps were plotted using the R package "pheatmap" v1.0.12 (https://github.com/raivokolde/pheatmap). Gene set enrichment analysis (GSEA) was performed using a plugin in TBtools, and the results were plotted using R (43).

For functional enrichment, GO and KEGG annotations were assigned using EggNOG-mapper. Significant enrichment of GO terms and KEGG pathways among DEGs was determined using TBtools with a background set of all annotated protein-coding genes (*P* < 0.05). The background gene set was defined as all protein-coding genes of the protist species annotated with KEGG pathways (https://www.genome.jp/kegg/tables/br08606.html#4).

## Analysis of alternative splicing

Alternative splicing (AS) events were identified using a comparative transcriptome assembly approach. Briefly, StringTie (v2.2.1) (44) was used to assemble transcripts independently for each sample, guided by the reference genome. The resulting individual transcriptome files (in GTF format) were then merged using the StringTie --merge command to generate a comprehensive, non-redundant set of transcript models. This merged transcriptome was compared to the reference annotation using gffcompare v0.12.6 (default thresholds, -e 100 -d 100) to classify the novel

and known transcripts (45). Finally, transcript classification codes (https://ccb.jhu.edu/software/stringtie/gffcompare.shtml) were used to categorize AS events into six types: intron compatible, retained introns (all), retained introns (partial), multi-exon, other same strand, and exonic overlap (opposite strand). The abundance of these AS events was then quantified from the merged GTF file using R. Statistical significance between different groups was analyzed using the $t$-test, with significance levels denoted as follows: $P < 0.05$ (*) and $P < 0.01$ (**).

## Ortholog detection and phylogenetic analysis

We identified the common orthologs among the 32 species using Orthofinder v2.5.4 (-S diamond -M msa -T raxml -I 1.5) (46). A maximum likelihood (ML) tree was constructed using RAxML-HPC2 (47). The ultrametric phylogenetic tree was constructed using r8s v1.81 (48), based on the rooted species tree inferred by OrthoFinder. Tree visualization was performed using MEGA v7.0.20 (49). Significantly expanded or contracted gene families were identified using CAFE (Computational Analysis of Gene Family Evolution) v5.0 with the parameter -k 5 (50). Thirty-one reference genomes were downloaded from public genome databases (Table S1). The amino acid sequences of the putative adenine MTase (AMT) family in *O. granulifera* were aligned with the MT-A70 protein in *Tetrahymena thermophila* using BLAST v2.9.0+ (29). The conserved domains of the MT-A70 family were identified using NCBI CD-Search (51). The accession numbers of genes used for phylogenetic analysis are shown in Table S2.

## 4D label-free proteomics

The samples were lysed using SDT (4% SDS, 100 mM Tris-HCl, 1 mM DTT, pH 7.6) lysis buffer for protein extraction. The amount of protein was quantified with the BCA Protein Assay Kit (Bio-Rad, USA). The protein suspensions were digested with trypsin overnight at 37°C. The peptides of each sample were desalted on C18 Cartridges (Empore SPE Cartridges C18 [standard density], bed I.D. 7 mm, volume 3 mL, Sigma), concentrated by vacuum centrifugation, and reconstituted in 40 µL of 0.1% (vol/vol) formic acid. Samples were separated by using the Easy-nLC system. Buffer A is 0.1% formic acid, and buffer B is 84% acetonitrile and 0.1% formic acid. Samples were loaded onto a reverse-phase trap column (Thermo Scientific Acclaim PepMap100, 100 µm × 2 cm, nanoViper C18) and separated with the C18 reversed-phase analytical column (Thermo Scientific Easy Column, 10 cm long, 75 µm inner diameter, 3 µm resin) at a flow rate of 300 nL/min controlled by IntelliFlow technology. For LC-MS/MS analysis, we employed 4D label-free proteomics, a high-resolution quantitative technique that combines liquid chromatography with trapped ion mobility spectrometry (TIMS), to add an additional separation dimension (mobility) to the traditional three dimensions (retention time, $m/z$, and intensity), thereby enhancing peptide identification sensitivity and throughput. Specifically, peptide samples were separated using an Easy-nLC system coupled to a timsTOF Pro mass spectrometer (Bruker, Germany). The mass spectrometry scan range was set to 100–1,700 $m/z$. Data acquisition was conducted using the parallel accumulation serial fragmentation (PASEF) mode. The 10-fold PASEF mode acquisition of mother ions was performed after first-level mass spectrum acquisition, using a cycle window time of 1.17 s. For secondary spectra with charge numbers of 0 to 5, the dynamic exclusion was set at 24 s. The MS raw data for each sample were combined and analyzed using MaxQuant 1.6.14 for identification and quantitation.

## Gene expression validation by RT-qPCR

The reaction of qRT-PCR was performed with a QuantStudio 5 instrument (Thermo Fisher Scientific, USA). For each sample, reactions were performed with three technical replicates. The qRT-PCR was carried out in a 20 µL reaction system containing 10 µL of 2× ChamQ Universal SYBR qPCR Master Mix (Vazyme, Q711-02), 0.2 µmol/L each of forward and reverse primers (added as 0.4 µL of 10 µmol/L stock solutions), 1 µL of cDNA

template, and nuclease-free ddH$_2$O to the final volume. The reaction procedures were as follows: (i) holding stage, predenaturation at 95°C for 30 s; (ii) PCR stage, 40 cycles were performed at 95°C for 3 s and 60°C for 30 s; and (iii) melt curve stage, 95°C for 15 s, 60°C for 60 s, and 95°C for 15 s. The primer sequences and gene information are listed in Table S3. All of the reactions were repeated in triplicate, and the relative expression levels were calculated using the $2^{-\Delta\Delta Ct}$ method (52).

## RESULTS

### Morphological transition of *Oxytricha granulifera* from the vegetative cells to cysts

Cells in the vegetative stage of *O. granulifera* were long ellipsoidal in shape and measured 80–120 µm long and 30–50 µm wide, with two large macronuclei and two or three micronuclei ($n = 25$) (Fig. 1A, C, and E). Mucocysts were ellipsoidal, with a length of about 0.5 µm, and were distributed longitudinally and irregularly on both the dorsal and ventral surfaces. Their total count was estimated to be 4,500–5,000 per individual ($n = 3$); their positive staining with FITC-Con A indicates their glycoprotein component since Con A (Fig. 1D and E). The adoral zone of membranelles (AZM) occupied about one-third of body length. Frontal, ventral, and transverse cirri were arranged in the typical 8-5-5 pattern. The left and right marginal rows were gently curved at the posterior end and composed of 23–27 and 23–31 cirri, respectively (Fig. 1C; Fig. S1A). In the TEM sections, most mucocysts were located beneath the pellicle and were elliptical in shape, with an electron-light cavity in the anterior part; mitochondria and starch granules were irregularly dispersed in the cytoplasm (Fig. 1H and I).

In the early stage of encystment, the body shortened posteriorly, both marginal rows became distinctively curved, and the transverse cirri and possibly the pre-transverse ventral cirri dedifferentiated; the proportion of the AZM to body length increased, but the cirri in the frontal area did not change (Fig. S1B). Later, the body shape was more oval (AZM occupied about 2/3 of body length), the number of cirri in the left and right marginal rows decreased to about half of their original number due to dedifferentiation from posterior to front; transverse cirri and pre-transverse ventral cirri disappeared completely, and postoral ventral cirri began to dedifferentiate (Fig. S1C). The de-differentiation of buccal apparatus began with the fragment of AZM, i.e., the membra-nelles in posterior part separated from adoral zone and de-differentiated (Fig. S1D and E). At this time, the posterior frontal-ventral cirri disappeared. Meanwhile, the left marginal row completely disappeared, and the right marginal row contained only several cirri (Fig. S1E).

Mature cysts were spherical and 30–50 µm in diameter, containing FITC-Con A–positive vesicles (likely mucocysts in autophagic vacuole) (Fig. 1B and G). Ultrastructural analysis of 20 TEM sections revealed two distinct layers in the immature cyst wall: a highly electron-dense outer layer (0.4–0.6 µm thick) and an uneven, electron-lucent inner layer (0.2–2 µm thick). A representative image is shown in Fig. 1J. The nucleoli in the fused macronucleus were more homogeneous and arranged more tightly compared to those of encysting cell (Fig. 1F and J). No kinetosomes or cilia remained in the cell cortex, while the starch granules and spherical electron-dense granules densely arranged near the cortex (0.1–0.2 µm in diameter) (Fig. 1J and K). A few autophagy vesicles containing mucocysts, starch granules, and mitochondria were also observed (Fig. 1L and M).

### Genomic features and evolutionary analysis in *Oxytricha granulifera*

Paired-end reads sequencing reads (15 Gb in total) were acquired from *O. granulifera* macronuclear genomic DNA, and the genome was assembled using SPAdes (Fig. 2A). Following the removal of bacterial and mitochondrial sequences, we performed deduplication on the genome assembly using a 98% sequence similarity cutoff. Subsequently, contigs with low support (coverage < 5 or length < 400 bp) were discarded. The final information of the genome was evaluated using QUAST (Fig. 2B). This assembly

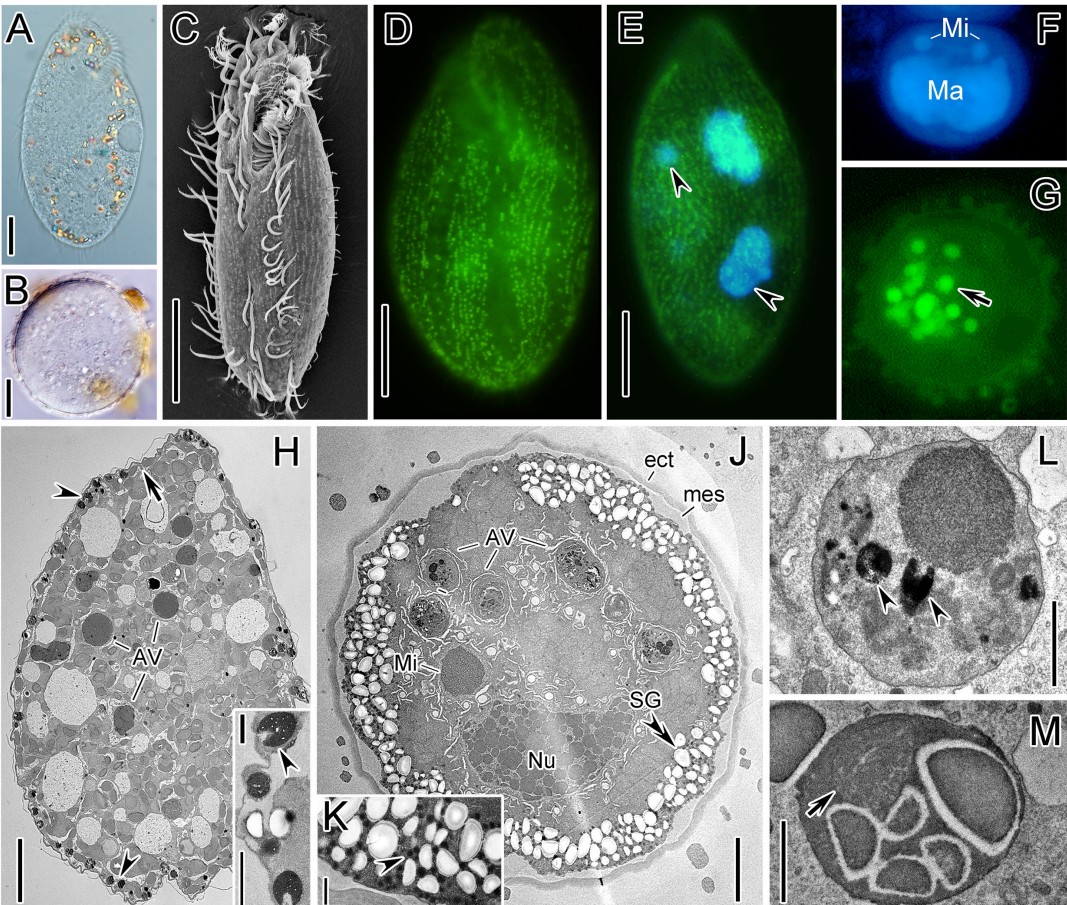

**FIG 1** Major morphological changes during encystment of *Oxytricha granulifera*. (A) Representative DIC image of vegetative cells under normal culture conditions. (B) Representative DIC image of mature cysts after 30 days of starvation. (C) SEM image of vegetative cell. (D, E) FITC-Con A fluorescence staining and DAPI fluorescence staining of the vegetative cell, showing the longitudinally arranged mucocysts distributed on both sides (green) and two macronuclei (blue). Arrowheads mark the micronuclei, which are usually associated with each macronucleus. (F, G) Mature cyst stained with DAPI (F) and FITC-Con A (G), showing the fused macronucleus (Ma), and the autophagic vacuoles (arrow) that enclose mucocysts. (H, I) Encysting cells without a cyst wall, showing the mitochondria (arrow) and mucocysts (arrowheads) beneath the pellicle. (J–M) Encysting cells with cyst wall, showing a large number of granules (likely the cyst wall precursors) (arrowhead) and starch granules (double arrowhead) near the cortex, the ectocyst (ect) and mesocyst (mes) of the cyst wall, the nucleoli of the fused macronuclei, and the homologous nucleoli (Mi), and autophagic vacuoles enclosing mucocysts (arrowheads) and mitochondria (arrow), as well as the endoplasmic reticulum around the autophagic vacuoles (double arrowhead). AV, autophagy vacuoles; ect, ectocyst; Ma, macronucleus; mes, mesocyst; Mi, micronucleus; Nu, nucleolus; SG, starch granule. Scale bars: 10 µm (A, B), 20 µm (C−E), 5 µm (H, J), and 1 µm (I, K, L, M).

comprised 16,944 contigs (41.7 Mb), with an N50 of 3,112 bp. The chromosomes of *O. granulifera* are approximately 2,000 bp in length, with most contigs containing one or two genes. This represents a characteristic genomic architecture shared with other spirotrichs (*Oxytricha trifallax*, *Euplotes vannus*, *Strombidium stylifer*) (Fig. 2B through D). Nanochromosomes usually carry single genes and have conserved potential cis-regulatory elements (CREs) in the 5′ subtelomeric regions, which regulate gene expression independently of one another (53). The GC content of the genome was 33.43% (Fig. 2B). EuGene predicted a total of 22,248 genes, with their sizes largely concentrated between 2,000 and 3,000 bp (Fig. 2B). As is typical for spirotrichs, the *O. granulifera* genome features nanochromosomes, elevated GC content, high gene density, and a low proportion of repetitive sequences. The telomeres of *O. granulifera* had the sequence motif of $[C_4A_4]_n$ (Fig. 2E). A canonical boundary motif of "GT-AT" was present in most predicted gene introns, which were short in size (20–30 nt) (Fig. 2F), and most genes had one exon (Fig. 2G). Genome assembly was assessed by BUSCO, with detection of 86.5% complete and 5.3% fragmented conserved orthologs (Fig. 2H).

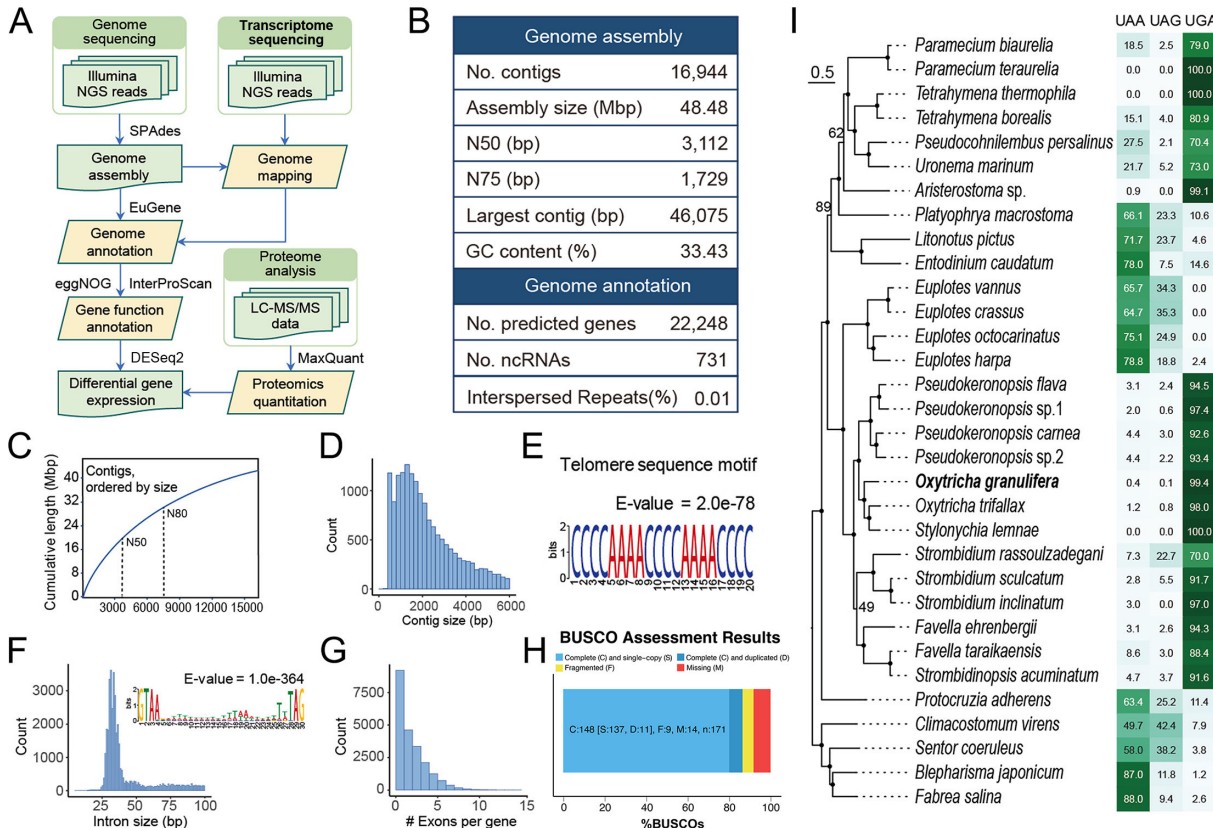

**FIG 2** Sequencing and assembly features of the somatic genome of *Oxytricha granulifera*. (A) Schematic diagram of the strategy for genome assembly and annotation. (B) Genome assembly features. (C) Distribution of cumulative length of the genome assembly. The N50 value represents the contig length at which 50% of the assembled genome is covered by contigs of this length or longer, while the N80 value indicates the contig length covering 80% of the genome. (D) Contig size distribution across the genome assembly. (E) Potential telomere sequence motif of the genome assembly. (F) Distribution of intron sizes and the sequence motif of the intron between 20 and 30 bp. (G) Exon distribution within genes. (H) BUSCO assessment of genome assembly completeness. The results are categorized as complete (C), fragmented (F), and missing (M) BUSCOs, providing a quantitative measure of how many of the expected core genes are fully present, partially represented, or absent from the assembly, respectively. (I) Phylogenetic tree based on orthofinder by maximum likelihood (ML) method. The scale bar corresponds to 30 substitutions per 100 positions. The adjacent heatmap to the right of the phylogenetic tree illustrates the utilization of stop codons across all 32 species of ciliates.

Ciliates often employ standard stop codons in unconventional ways, diverging from the typical usage observed in other eukaryotic organisms (54). Genome assembly and gene structure annotation strongly suggest that *O. granulifera*, like other spirotrichs, uses UGA as the sole stop codon and reassigns UAA and UAG to encode amino acids (Fig. 2I). In the close relative *Oxytricha trifallax*, these codons specify glutamine (55). Given the high genomic conservation between these species, we infer that the UAR codons (UAA and UAG) in *O. granulifera* also encode glutamine.

To elucidate the evolutionary characteristics of gene families in *O. granulifera* and identify the molecular basis underlying its encystment ability, we collected predicted protein sequences of 32 species and performed homologous gene clustering analysis on 762,541 protein-coding genes using OrthoFinder. The results showed that a total of 88,482 orthologous gene families (Orthogroups) were identified, among which 639,495 genes (83.9% of the total genes) were successfully assigned to each family. Among all identified homologous families, 36,300 were species-specific orthogroups, comprising 135,585 genes (17.8% of the total genes). This high proportion of species-specific orthogroups is expected for such distantly related taxa and likely reflects limited homology detection due to significant sequence divergence rather than extensive *de novo* gene origination (Table S4). Subsequently, we analyzed the expanded and

contracted orthogroups in *O. granulifera* and identified 468 orthogroups with significant expansion and 907 orthogroups with significant contraction (Fig. 3A). GO enrichment analysis identified 155 significantly enriched GO terms ($P < 0.05$) in *O. granulifera*, with the top 20 terms—ranked by *q*-values—covering all three aspects: molecular functions, biological processes, and cellular components (Fig. 3B). They were predominantly associated with membrane components, transmembrane transport, peptidase activity, proteolysis, and carbohydrate metabolism, which confer *O. granulifera* enhanced adaptive capabilities in nutrient acquisition, energy homeostasis, and environmental stress resistance. The extensive presence of mucopolysaccharides or glycoproteins, characterized as α-mannose in the mucocysts distributed across the cortex of *O. granulifera*, potentially correlates with the gene expansion (particularly mannose metabolic process and hydrolase activity) detected in this species.

Additionally, we identified multiple potential encystment-associated terms among the significantly enriched GO terms in expanded gene families of *O. granulifera* (Fig. 3C), according to previous studies of ciliate gene family expansion, encystment mechanisms, and encystment-related transcriptomes (e.g., references 17, 27). Among these, negative regulation of Target of Rapamycin Complex 1 (TORC1) signaling, cellular response to environmental stimuli, response to water deprivation, response to calcium ions, and monoatomic ion transport are strongly linked to the process of encystation (Fig. 3C).

## Transcriptome and proteome analyses revealing the genetic basis during cyst formation of *O. granulifera*

To uncover the transcriptional profiles associated with the encystment of *O. granulifera*, we compared vegetative cells and resting cysts of *O. granulifera* via RNA-seq (three replicates per stage) to identify differentially expressed genes. After normalizing the read counts, the similarity between samples from different tissue types was evaluated by generating a principal component analysis (PCA) plot using the normalized count data. PCA of the data set showed that the samples segregated primarily based on the cyst stage (Fig. 4A). Transcriptome profiles showed a dramatic change in cyst stage cells compared to those in vegetative stage (Fig. 4B). To identify differentially expressed genes (DEGs; fold change > 2 and *q*-value < 0.1), we generated a volcano plot. This analysis revealed 13,810 DEGs, comprising 9,967 downregulated and 3,843 upregulated genes (Fig. 4C; Table S5). To validate the results of transcriptome analyses, five genes were selected to confirm the expression level using RT-qPCR. The fold change values derived from RT-qPCR and RNA-seq expression data were predominantly in agreement (Fig. S2D).

We performed GO enrichment analysis for the DEGs. The upregulated genes were enriched in U5 snRNP, spliceosomal complex, U2-type spliceosomal complex, ubiquitin protein ligase activity, oxidoreductase activity, and ubiquitin-protein transferase activity (Fig. 4D; Table S6), while the downregulated genes were enriched in exopeptidase activity, protein metabolic process, cytosolic ribosome, and amide metabolic process (Fig. S2A; Table S6). The GSEA analysis reveals similar results in GO enrichment. Notable upregulated processes include "U2 type spliceosomal complexes" and "Regulation of RNA splicing," while the downregulated processes include "Regulation of cellular response to growth factor stimulus" (Fig. 4E; Fig. S2E). A GSEA plot shows the enrichment of the U2-type spliceosomal complex (GO:0005684, CC) in the ranked gene list (Fig. 4F). The red line represents the enrichment score (ES), which peaks at 0.7486, indicating significant enrichment. The normalized enrichment score (NES) is 2.6192, which further highlights the significant upregulation of this term. The KEGG analysis based on the transcriptomic data set showed a similar result, with upregulated genes enriched in spliceosome, replication and repair, protein export, peroxisome, and DNA replication (Fig. S2B; Table S6). The downregulated genes were enriched in cytoskeleton proteins, glycan biosynthesis, and protein kinases (Fig. S2C; Table S6).

In our study, a significant downregulation was observed in members of the intraflagellar transport (*IFT*) and tubulin tyrosine ligase-like (*TTLL*) families among the differentially expressed genes. Specifically, the expression levels of IFT-A complex proteins

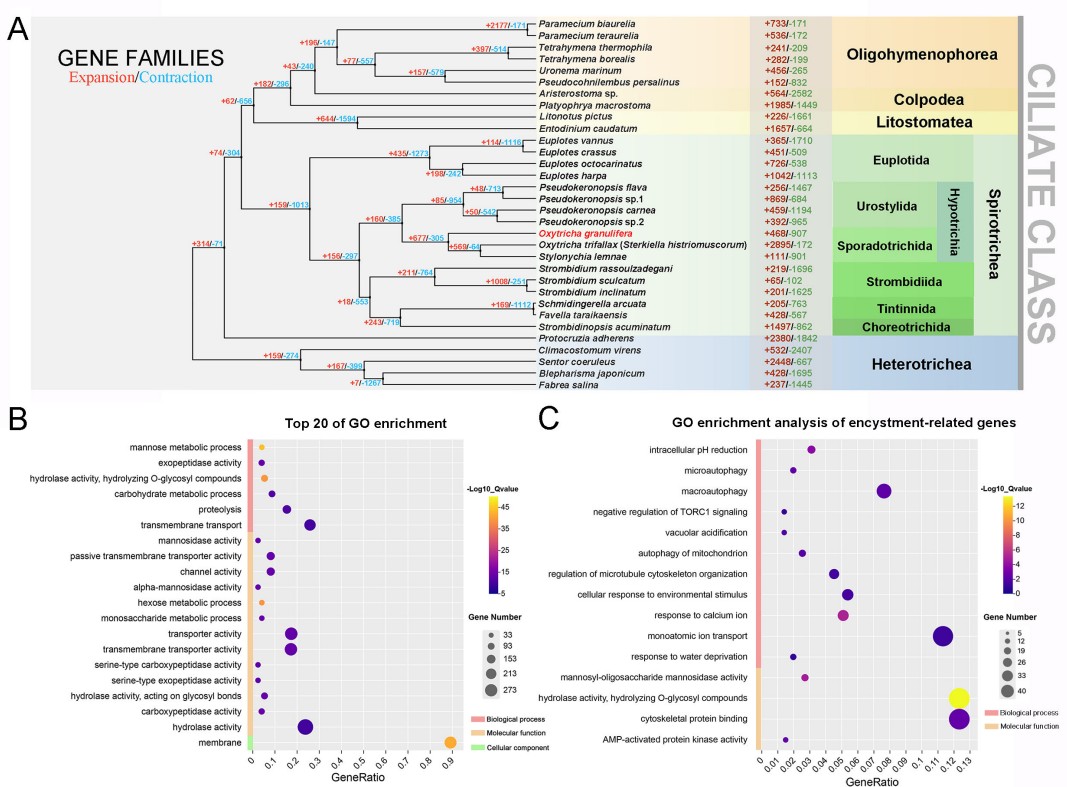

**FIG 3** Analyses of evolutionary history and gene family expansion and contraction of *Oxytricha granulifera*. (A) The phylogenomic tree, gene family expansion, and contraction for *O. granulifera* and 31 other species. The numbers at nodes indicate the number of gene families expanded and contracted at different evolutionary time points. Numbers following species names represent expanded and contracted gene families for that species. (B and C) The top 20 GO terms (B) and encystment-related GO terms (C) in significantly expanded gene families in *O. granulifera*. GO, Gene Ontology; GeneRatio, the ratio of enriched gene number to all gene number in this pathway term.

(*IFT122*) and IFT-B complex proteins (*IFT25* and *IFT22*) were markedly reduced. Similarly, several *TTLL* family members—including *TTLL2*, *TTLL4*, *TTLL10*, and *TTLL11*—also exhibited decreased expression (Fig. 4C), suggesting potential impairment in tubulin modification dynamics (25). Conversely, we observed upregulation of vesicular transport machinery, including coatomer and adaptin complexes. Furthermore, both the catalytic α subunit (*PRKAA1*) and regulatory β subunits (*PRKAB1/PRKAB2*) of AMPK showed concurrent upregulation.

In order to elucidate the regulatory mechanisms governing gene expression during the encystment process, we further performed a comprehensive proteomic analysis. PCA analysis was employed to illustrate the correlations among samples, revealing a high degree of similarity within both the cyst and veg group (Fig. 5A). Similar to the case in the transcriptome (Fig. 4B), cells in the cyst stage had dramatically shifted proteome profiles from those in the vegetative stage (Fig. 5B). In the proteomics data, we identified 346 upregulated genes and 369 downregulated genes during encystment (fold change > 1.5 and *P*-value < 0.05; Fig. 5C; Table S5). The KEGG pathway enrichment analysis showed that the upregulated proteins were related to proteasome, spliceosome, and exosome (Fig. 5D). The GO enrichment analysis showed that the upregulated proteins were enriched in the proteasome and peptide metabolic process, while the downregulated proteins were enriched in lipid metabolism and amino acid metabolism (specifically, glycine, serine, and threonine metabolism) (Fig. 5E; Fig. S3, Table S7).

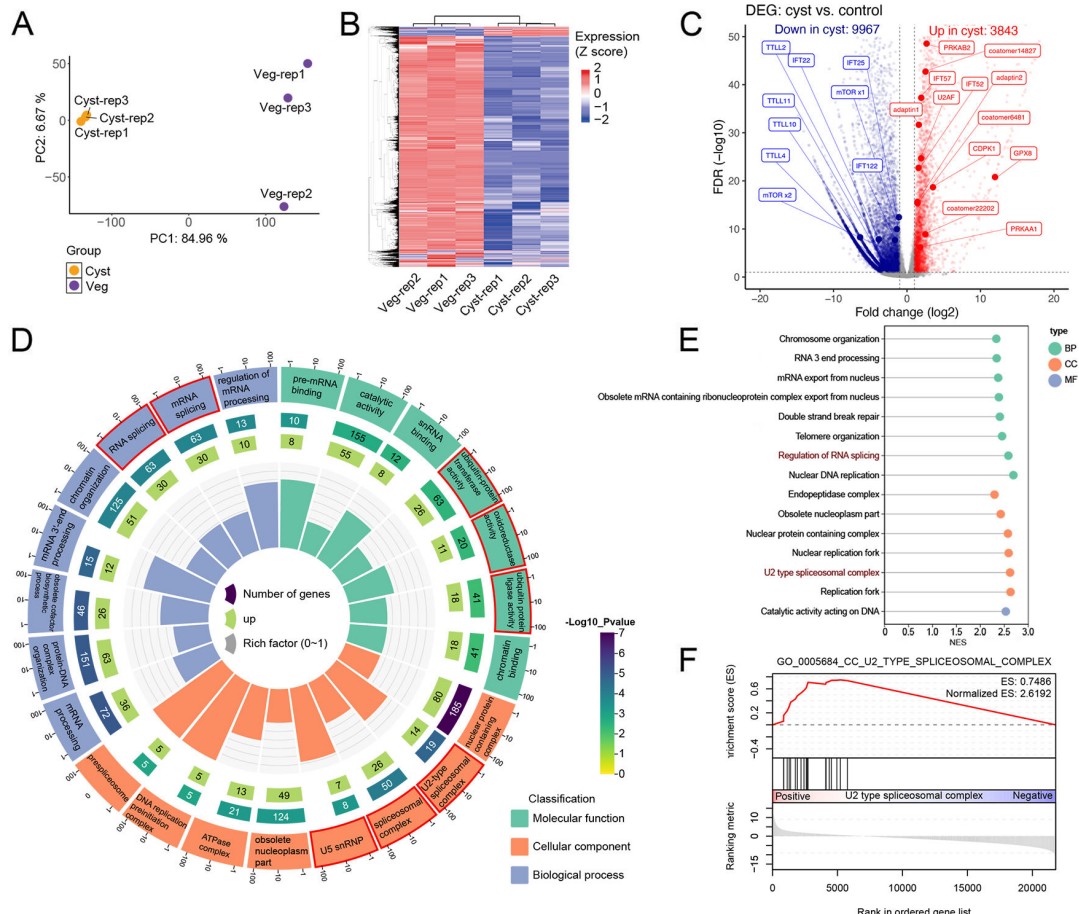

**FIG 4** Gene expression changes between vegetative and cyst stages of *Oxytricha granulifera*. (A) Principal component analysis (PCA) based on gene expression profiles. Each point represents a sample. Orange dots represent the vegetative stage (Veg), while purple dots represent the cyst stage (Cyst). (B) Heatmap of global gene expression patterns across different stages. (C) Volcano plot of gene expression with fold change thresholds of 2 and an adjusted *P*-value threshold of 0.1. (D) GO circle plot. GO circle plot shows the enrichment of molecular functions, biological processes, and cellular components. The inner ring is a bar plot where the height of the bar indicates the enrichment factor of the term, and the color indicates GO terms with which the corresponding genes are associated. The outer ring displays the upregulated genes in each term. The third ring shows the total number of genes associated with this term, and the color corresponds to the significance ($-\log_{10}$ *P*-value). The outermost ring displays the names of the GO term. The enriched term associated with encystation is highlighted in red. (E) Results of gene set enrichment analysis (GSEA). Normalized enrichment scores were shown on the *x*-axis. The *y*-axis lists the enriched biological processes and molecular functions, categorized by Gene Ontology (GO) annotations into molecular function (MF; green), cellular component (CC; orange), and biological process (BP; blue). (F) GSEA analysis of the U2-type spliceosomal complex.

## Identifying key functional molecular events during cyst formation of *O. granulifera*

We compared the abundance of seven genes that play important roles in encystment between RNA and protein levels. Transcriptome and proteome analyses confirmed the high expression of these proteins during the encystment period (Fig. 5F). We observed a significant upregulation of UFD1 (ubiquitin folding domain containing 1), SOD1 (superoxide dismutase 1), PSMC6 (proteasome 26S subunit, ATPase, 6), CTSB (cathepsin B), α-GalA (alpha-galactosidase A), SNRNP40 (small nuclear ribonucleoprotein polypeptide 40), RPN9 (ribophorin 9), and NFS1 (NAD(P)H:quinone oxidoreductase 1). UFD1, PSMC6, CTSB, and RPN9 were involved in the intracellular degradation of proteins, supporting the cell's transition into the encysted state. SOD1 and NFS1 may play a key role in protecting cells from ROS (26, 56). SNRNP40 participates in the biogenesis of small nuclear ribonucleoproteins (snRNPs), potentially influencing the splicing and maturation of pre-mRNA (57). α-Gal A may be involved in the metabolism of

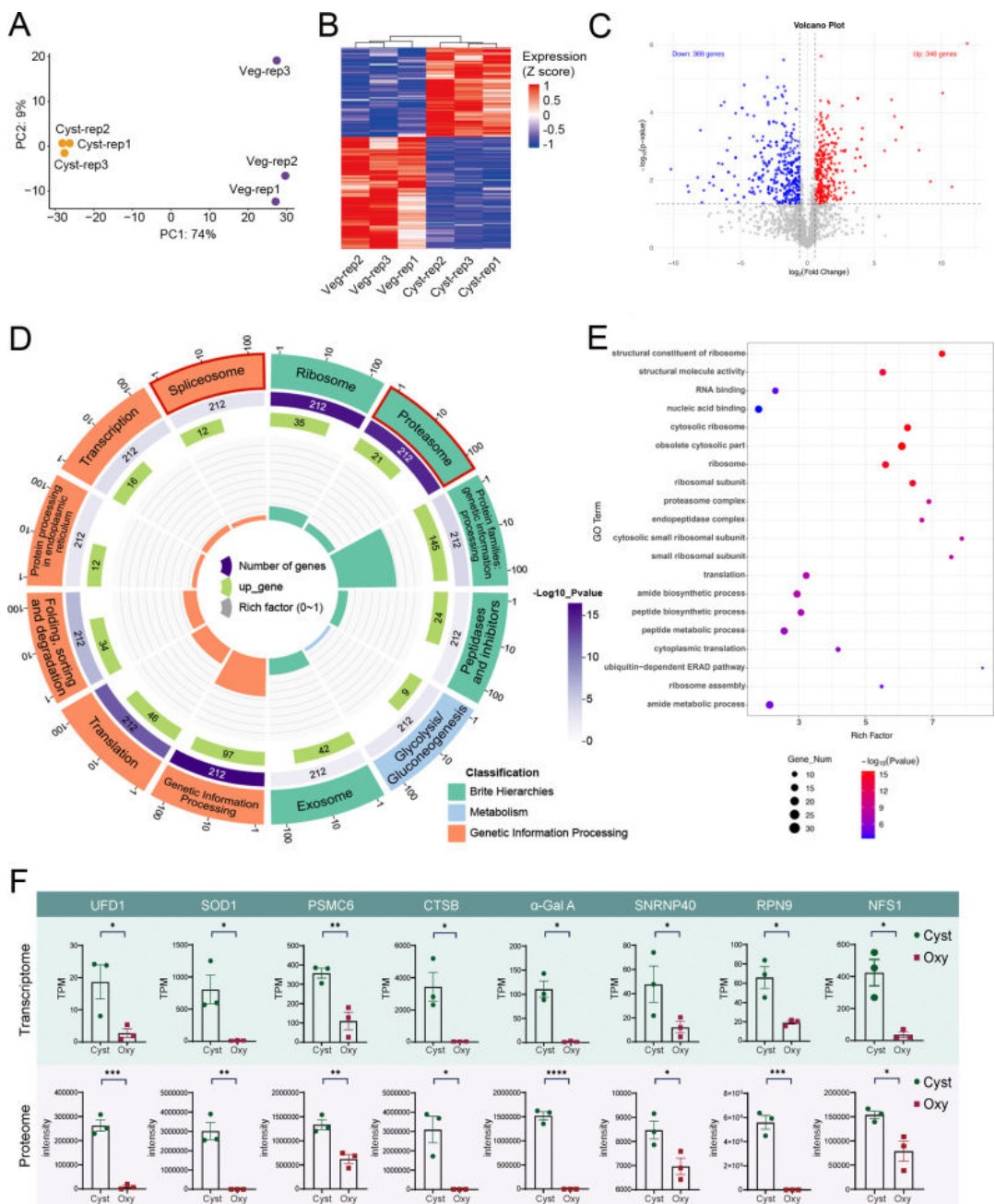

**FIG 5** Protein changes between vegetative and cyst stages of *Oxytricha granulifera*. (A) Principal component analysis (PCA) based on proteome profiles. Each point represents a sample. Orange dots represent the vegetative stage, while purple dots represent the cyst stage. (B) Heatmap of proteomic across different stages. (C) Volcano plot of proteomic with fold change thresholds of 1.5 and a *P*-value threshold of 0.05. (D) KEGG circle plot. The inner ring is a bar plot where the height of the bar indicates the enrich factor of the term, and the color indicates KEGG terms with which the corresponding proteins are associated. The outer ring displays the upregulated proteins in each term. The third ring shows the total number of proteins associated with each term, and color corresponds to the significance (−log$_{10}$ *P*-value). The outermost ring displays the names of the KEGG terms. The enriched term associated with spliceosome is highlighted in red. (E) GO enrichment. (F) Plots showing RNA and protein (bottom) abundance of key genes. RNA abundance is measured as transcript per million (TPM). Significance levels denoted as follows: $P < 0.05$ (*), $P < 0.01$ (**), $P < 0.001$ (***) and $P < 0.0001$ (****).

cell surface glycoproteins and glycolipids, thereby affecting cell adhesion and signal transduction (58, 59). These findings collectively reinforce the reliability of our differential gene expression analyses based on RNA-seq. Furthermore, these multi-omics reveal that the coordinated upregulation of proteostasis and downregulation of core

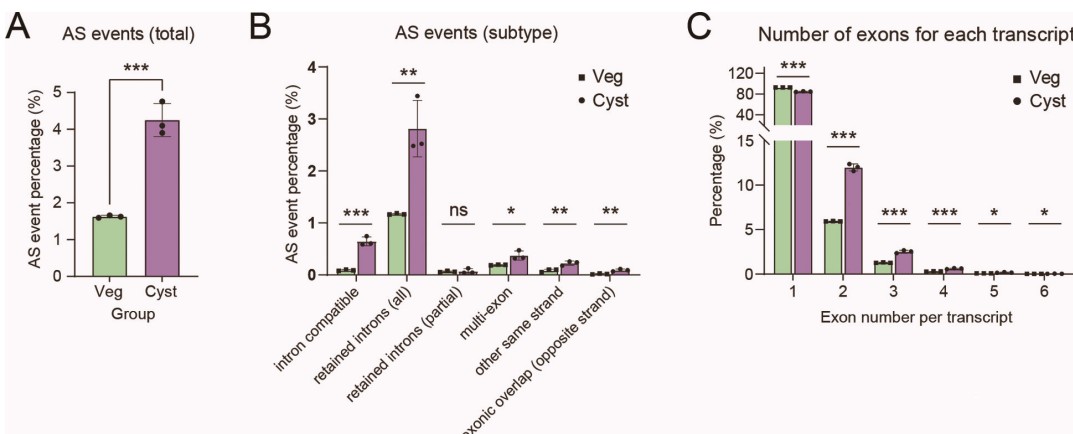

**FIG 6** Alternative splicing analysis during the vegetative cell-to-cyst transition of *Oxytricha granulifera*. (A) Relative frequencies of alternative splicing events compared to the total number of total transcripts in cells at the vegetative stage (Veg) and cyst stage (Cyst). (B) Subtypes of alternative splicing events between vegetative stage (Veg) and cyst stage (Cyst). (C) Number of exons for each transcript between two groups. *, $P < 0.05$; **, $P < 0.01$; ***, $P < 0.001$.

metabolism facilitate rapid encystment, a molecular adaptation critical for surviving sudden environmental adversity.

It is worth noting that, in both transcriptome and proteome data, we observed upregulation of several genes or proteins associated with spliceosome, indicating the presence of numerous alternative splicing events during encystment (Fig. 4D and 5D). Considering the global transcription-level difference between encysting and vegetative cells, we analyzed the frequency of alternative splicing events relative to the number of total transcripts, and the results indicated that alternative splicing occurred more frequently during encystment (Fig. 6A). Regarding the types of alternative splicing, there were slightly more events during encystment than during the vegetative stage, primarily involving intron compatible, retained introns, and other types of alternative splicing (Fig. 6B). We also examined the number of exons per transcript, and the results showed that the majority of transcripts during both stages had one exon (Fig. 6C). However, transcripts with two exons were significantly more common during encystment, possibly reflecting the gene expression characteristics during this stage.

DNA $N^6$-adenine methylation (6mA) is a transcriptional activator involved in gene regulation and is catalyzed by 6mA methyltransferase (6mA MTase). An expanding list of studies shows that 6mA responds to external stress. For example, in the worm *Caenorhabditis elegans*, 6mA is involved in mitochondrial stress adaptation (60). In addition, in *Pseudocohnilembus persalinus*, the methyltransferase PpAMT1 regulates cellular growth and encystment by modulating 6mA levels, a conserved regulatory mechanism among ciliates. Reducing the levels of AMT enzymes accelerates encystment, highlighting the critical role of these enzymes in controlling the encystment process (20).

In this study, we identified the methyltransferases responsible for 6mA methylation in *O. granulifera*, confirmed the conserved domains of the AMT family using NCBI CD-Search (Fig. 7A), and reconstructed the phylogenetic tree of AMT family sequences from representative eukaryotic species (Fig. 7B). Similar to *P. persalinus*, we found that the homologous genes of AMT1 and AMT6/7 in *O. granulifera* are also significantly downregulated during encystment (Fig. 7C). This suggests that 6mA may have a similar negative regulatory role during encystment in *O. granulifera*.

## DISCUSSION

The formation of dormant cysts involves complex and highly regulated physiological and biochemical alterations. Through comprehensive analyses of morphology, gene family expansion, transcriptomics, and proteomics in *O. granulifera*, we investigate how cellular stress responses trigger an integrated signaling network and coordinate

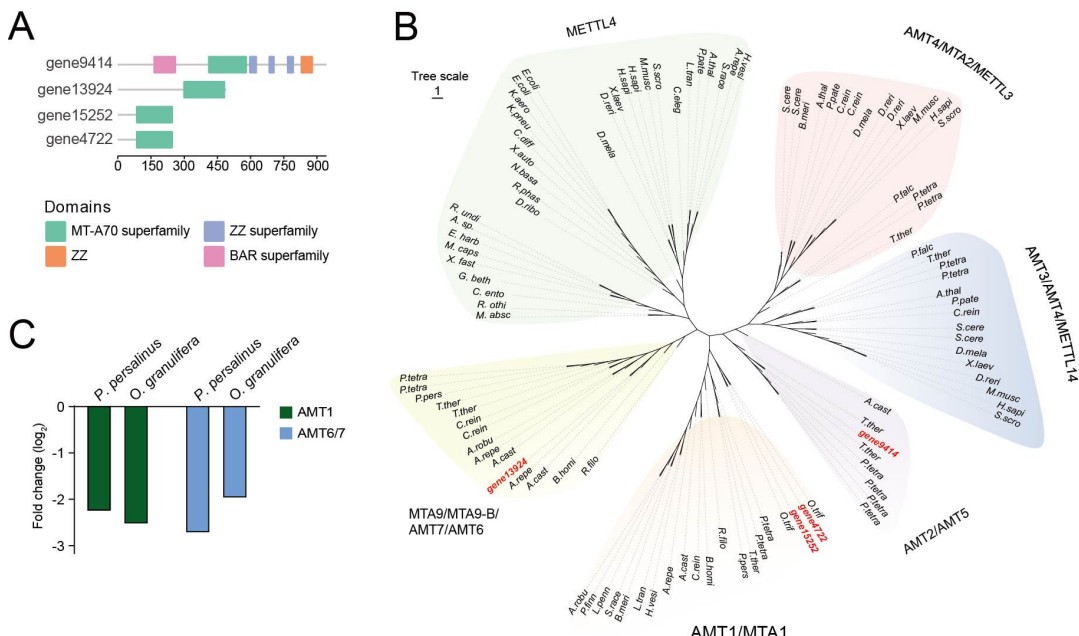

**FIG 7** Phylogenetic analysis and domain structure of OgAMT proteins and their differential expression ratios. (A) Domain structures of four potential OgAMT proteins. (B) Phylogenetic analysis of MT-A70 proteins. Full species names are listed in Table S2. (C) Gene expression levels of potential AMT proteins in *Pseudocohnilembus persalinus* and *Oxytricha granulifera*.

organelle-specific adaptations. We further elucidate how this network orchestrates the physiological changes essential for encystation.

## Signal transduction regulation

The initiation of ciliates' dormancy involves environmental signal perception and signal transduction, and the elevated intracellular $Ca^{2+}$ concentrations play the core role by modulating intracellular signaling pathways (16, 17, 61). In *O. granulifera*, we identified several expanded gene families that are significantly enriched in functions related to environmental stress response (e.g., "cellular response to environmental stimulus" and "response to water deprivation"), suggesting an enhanced capacity for environmental stress perception in this species. Correspondingly, the enrichment of "response to calcium ion" indicates a possible reinforced regulatory capacity of $Ca^{2+}$ signaling pathways. Additionally, the enrichment of "monoatomic ion transport" indicates the enhancement of the overall capacity of re-establishing cellular ion homeostasis throughout the encystation process, other than the rapid ion-mediated responses in the initial stage.

In *Pseudocohnilembus persalinus*, the dynamic 6mA modification exhibits a transition from symmetric to asymmetric sites during the vegetative cell-to-cyst transformation, which is closely associated with gene expression regulation, and a reduction in 6mA levels facilitates cyst formation (20). Calcium-dependent protein kinase (CDPK) is the most abundant class of calcium sensors, being found in protozoa, ciliates, and plants (62). Emerging evidence suggests their potential functional importance in *Cryptosporidium parvum* (an apicomplexan) growth (63). In *O. granulifera*, transcriptomic analysis revealed that both *CDPK* and 6mA-related enzyme genes were downregulated. Based on these expression patterns and domain conservation, we hypothesize that such downregulation may contribute to the inhibition of vegetative growth and the promotion of cyst formation, and that reduced 6mA levels may act as a potential regulatory signal in this process.

## RNA processing regulation

Alternative splicing (AS) is the process through which primary transcripts can be modified in different arrangements to produce functionally distinct mature mRNAs, and AS events are regulated to ensure the production of appropriate protein iso-forms in the correct cellular environments (64). Splicing is carried out in a stepwise, coordinated fashion by a large ribonucleoprotein complex, named spliceosome—a molecular machine composed of five ribonucleoprotein particles (snRNPs: U1, U2, U4, U5, and U6) (65). Our data show upregulation of the U2 snRNP spliceosome and U5 snRNP in transcriptomic GO enrichment, along with the spliceosome pathway in proteomic KEGG enrichment. Additionally, the upregulation of U2 small nuclear RNA auxiliary factor 1 (*U2AF*, U2 snRNP component for splice site recognition) and *SNRNP40* (core protein component of U5 snRNP) among differentially expressed genes (DEGs) suggests heightened spliceosomal activity during encystment formation. This coordinated induction of both early (U2-associated) and late (U5-associated) splicing factors may facilitate proteome remodeling through enhanced RNA processing, potentially regulating stage-specific physiological adaptations required for encystment. Alternative splicing is commonly classified into seven types of simple binary events, among which intron retention is the most prevalent mode of alternative splicing in unicellular eukaryotes. It may lead to alterations in the secondary structure of mRNA, thereby affecting ribosome binding and translation initiation, and ultimately reducing protein synthesis (66–69). Intron retention increases during cyst formation of *Acanthamoeba castellanii* (Amoebozoa), indicating a potential mechanism of gene regulation that could help downregulate metabolism (70, 71). In the study on yeast, intron retention has been linked to cellular responses to starvation and stress (72, 73). The enrichment of alternative splicing events among upregulated genes in *O. granulifera*, along with a high frequency of intron retention and intron-compatible alternative splicing, suggests that intron retention is likely a common mechanism for single unicellular eukaryotes to cope with stressful conditions.

## Intracellular resource repurposing

The formation of cysts and the regression of certain cellular structures require the involvement of autophagy and protein degradation pathways. The ubiquitin-proteasome system (UPS) is a crucial pathway for intracellular protein degradation, marking proteins for degradation through ubiquitination and then degrading them via the proteasome (74, 75). In our study, we observed an enrichment of the ubiquitin-related protein degradation pathway in both the transcriptome and proteome (e.g., UFD1, PSMC6, CTSB, and RPN9) (Fig. 5F). The expression of ubiquitin cascade components E2 and E3 in ubiquitin degradation signaling pathway was significantly upregulated, suggesting that the UPS is highly activated, which responds to the fast and efficient degradation of damaged and unneeded cellular proteins. PSMC6, also known as proteasome 26S subunit ATPase 6, is a component of the 19S regulatory particle of the 26S proteasome and plays a crucial role in the degradation of proteins marked with ubiquitin (17, 74, 76). The unnecessary cellular proteins are degraded, which meet the needs of the cell morphological upheaval and rapid cell shrinkage during the cyst formation. This mechanism is indispensable for the encystment process of ciliates, enabling their survival and adaptation to dynamic environmental challenges (17).

In ciliates, extrusomes exhibit diverse fates within cysts, distinct from their roles during the trophic phase. For instance, in *Colpoda cucullus* and *Tetrahymena rostrata*, mucocysts expel mucus to form the cyst wall (77, 78). In *Pseudourostyla cristata*, protrichocysts not only contribute to the cyst wall but also undergo autophagy (79). Autophagy is a process by which cellular proteins and damaged or excess organelles are degraded through the formation of a double-membrane structure known as the autophagosome (80, 81). The mTOR complex is an important regulatory molecule that inhibits autophagy, and its negative regulation promotes autophagy (82, 83). In *O. granulifera*, the presence of autophagic vesicles engulfing mucocysts and glycoproteins

within cysts suggests that mucocysts are degraded via autophagy rather than being secreted. This autophagic degradation of extrusomes may therefore represent an alternative pathway for cyst wall formation, where internal recycling of glycoprotein components necessitates enhanced catabolic and anabolic capacities (84). We speculate that the significant expansion of gene families related to carbohydrate metabolism may enable *O. granulifera* to utilize the glycoproteins in extrusome components not only as a sustained energy source during early encystment but also as potential materials for cyst wall constituents. The coordinated enrichment of "AMP-activated protein kinase activity" and "negative regulation of TORC1 signaling" suggests a potential enhancement of AMPK's capacity to activate autophagy, possibly by suppressing TORC1-mediated inhibition. The transcriptional upregulation of protein kinase AMP-activated catalytic subunit α (PRKAA) and protein kinase AMP-activated non-catalytic subunit β (PRKAB), which encode the α- and β-subunits of AMPK, respectively, leads to activation of the AMPK complex (85–87). Subsequently, activated AMPK phosphorylates downstream effectors, suppresses mTOR signaling, and thereby modulates autophagy, consistent with the observation of autophagic vacuoles encapsulating organelles and the expansion of gene families. Previous studies have shown that cyst cells maintain a low respiration rate, with mitochondria undergoing degradation or aggregation and exhibiting minimal activity during encystment (77). Most notably, the expansion of mitophagy-related genes and macroautophagy aligns with the frequent encapsulation of mitochondria within autophagic vacuoles (Fig. 1J and M). This multi-layered evidence highlights autophagy as a central mechanism for ciliate encystment, safeguarding cellular survival through efficient resource recycling and organelle quality control in a metabolically quiescent state.

In the dormant forms of other groups of single-celled eukaryotes, such as yeast spores (*Saccharomyces cerevisiae*) and resting cells of diatoms (e.g., *Thalassiosira pseudonana*), reduced intracellular pH is a characteristic feature of encystment, and this acidification contributes to the preservation of the native structure of proteins and their oligomeric states, as observed in yeast spores (88, 89). The enrichment of "intracellular pH reduction" and "vacuolar acidification" in gene expansion of *O. granulifera* may be associated with enhanced autophagy-mediated waste degradation, as well as reduced metabolism while maintaining the structure and function of important proteins.

## Mechanism of morphological remodeling

Morphologically, the encystment is characterized by the formation of cyst-specific structures and the dedifferentiation of vegetative motility and feeding structures. Oxytrichids usually have four layers (ectocyst, mesocyst, endocyst, and granular layer) in cyst wall (10). In our TEM observation, only two layers of the cyst wall were found in the encysting cell, and a large number of cortical electron-dense granules, which are the same size as the precursors of granular layer in other oxytrichids (10). Concurrently, the expression of coatomer- and adaptin-related genes was upregulated. Coatomer constitutes the structural protein core of the coat protein I (COPI) complex. The COPI complex transports vesicles backward from the Golgi to the ER and within the Golgi, keeping organelles balanced. Adapting complexes cooperate with clathrin to mediate endocytic and late secretory transport (90, 91). In *Colpoda steinii*, the cyst wall precursors—ER-derived small vesicles abundant during encystment but absent from vegetative cells—originate exclusively from the rough endoplasmic reticulum (92). It was thus speculated that upregulated vesicular transport-related genes are likely important for cyst wall formation by orchestrating the vesicular transport of the precursors.

Ciliature, as the major component of the tubulin cytoskeleton of ciliate cells, undergoes varying degrees of resorption during encystment in different ciliate groups, resulting in three artificially divided types of cysts: non-kinetosome-resorbing cysts (only the ciliary shafts were partially dedifferentiated); partial-kinetosome-resorbing cysts (most ciliature components dedifferentiated, while some kinetosomes remained intact); and kinetosome-resorbing cysts (all the cilia, kinetosomes, and microtubules

were absorbed) (6). Given the absence of ciliary components within the cysts, the cysts are of the kinetosome-resorbing type. The observed expansion of gene families associated with "regulation of microtubule cytoskeleton organization" and "cytoskeletal protein binding" suggests enhanced molecular capacity for both complete ciliary structure disassembly during encystment and efficient cytoskeletal reconstitution during excystation.

The downregulation of *TTLL* and *IFT* may help reprogram ciliature during encystment in *O. granulifera*. Ciliary assembly and maintenance are governed by bidirectional IFT, whose two principal subcomplexes—IFT-A and IFT-B—execute distinct cargo itineraries: IFT-A mediates retrograde trafficking of membrane proteins, whereas IFT-B orchestrates anterograde delivery of soluble precursors, including tubulin (93). The downregulation of genes encoding the bidirectional IFT machinery indicates that a precisely balanced IFT flux is indispensable for cilium biogenesis, structural homeostasis, and signal transduction. *TTLLs* orchestrate tubulin assembly, maintenance, and motility by encoding enzymes that post-translationally modify tubulin via polyglutamylation, polyglycylation, and tyrosination (94, 95). In the *P. cristata* cyst, the downregulation of *TTLL11* led to the speculation that the poor microtubule transport functions facilitate the entry into dormancy (27), whereas the identification of more TTLL family genes in *O. granulifera* further supports this idea. Although autophagy can dismantle cilia during the encystment of ciliates (6), our findings indicate that the primary mechanism is a transcriptional fine-tuning of cilia-dynamic–related genes, which facilitates an orderly disassembly of the ciliary structure rather than its wholesale degradation.

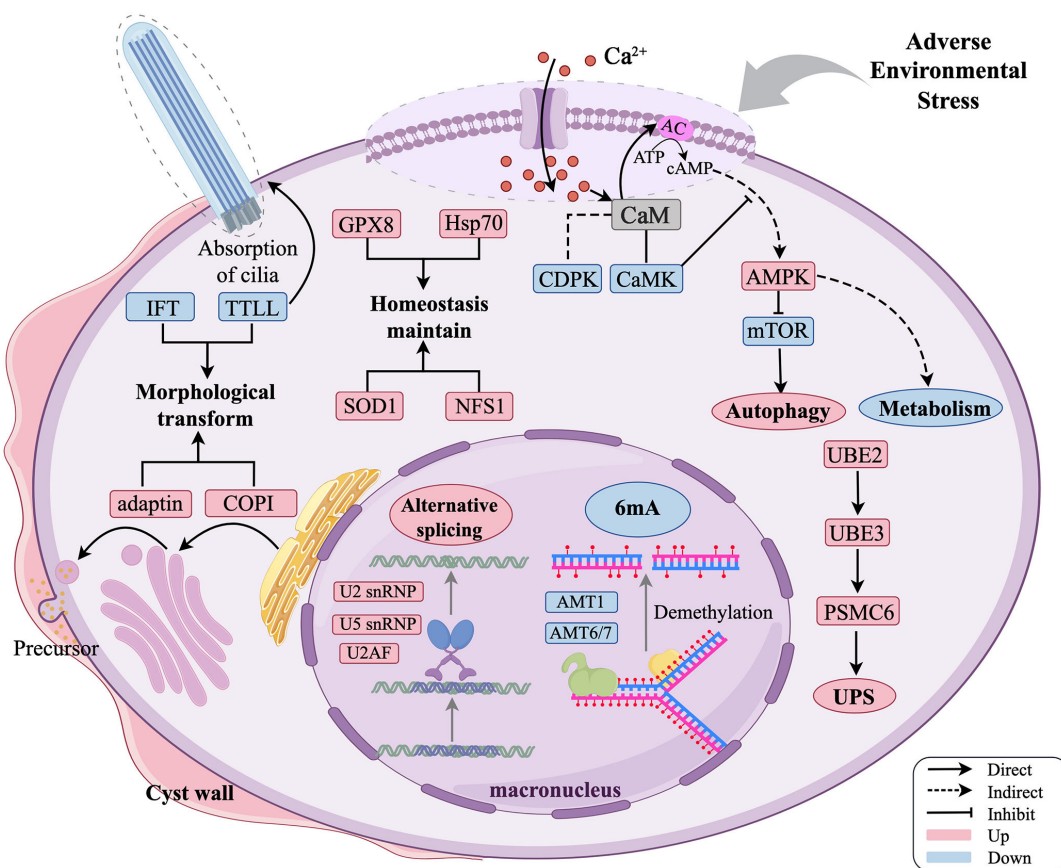

**FIG 8** Schematic diagram of major morphological changes and the hypothetical regulating signaling network during encystment of *Oxytricha granulifera*. Pink and blue colors indicate upregulation and downregulation of molecular events, respectively.

## Coordinated metabolic downscaling and stress defense

Downregulation of cytoskeleton- and motor protein-related genes and proteins induces structural alterations in cells, reduces cellular volume and water content, and leads to the stagnation of essential physiological processes such as locomotion and feeding. At the same time, our results demonstrate that pathways related to cell motility, metabolism, glutathione metabolism, and carbohydrate metabolism pathways were significantly downregulated according to KEGG enrichment analysis based on the proteomic data set (Fig. S1C and S3B). These findings indicate a metabolic shift toward energy conservation during the encystment process in *O. granulifera*. To ensure survival under this new, low-energy regime, a parallel enhancement of antioxidant defenses is critical.

Several proteins involved in oxidative stress response, including the molecular chaperone heat shock protein 70 (Hsp70) and glutathione peroxidase 8 (GPX8), were identified as regulators during encystment, primarily through their role in scavenging organic peroxides (18, 96). Our findings demonstrate significant upregulation of antioxidant proteins SOD1 and NFS1 in both transcriptomic and proteomic profiles of cysts, along with a marked transcriptional elevation of GPX8 (Fig. 5F). This suggests their synergistic role in protecting cells from oxidative stress by enhancing oxidoreductase activity. This enhanced antioxidant defense works synergistically with metabolic reprogramming, structural remodeling, and protein homeostasis to establish a new steady state during the transition from vegetative growth to encystment.

## Conclusion, limitations, and perspectives

In our study, we comprehensively investigated the encystment of *O. granulifera*, obtaining detailed morphological changes, assembling a relatively complete macronuclear genome sequence for the first time, and acquiring both transcriptomic and proteomic data across the vegetative and cyst stages. These findings provide a comprehensive understanding of the encystment process in ciliates, highlighting the coordinated changes at both morphological and molecular levels (Fig. 8). In conclusion, this work represents a significant advance in the research on dormancy of these understudied single-celled eukaryotes. It achieves this through the application of integrated "omics" tools and advanced microscopy techniques. Therefore, this work lays the foundation for a deeper understanding of the life history and environmental adaptations of soil ciliates.

Our study possesses certain limitations that should be considered. First, as with most mass spectrometry-based proteomics, our analysis likely does not capture the complete proteome. This incomplete coverage is influenced by technical factors such as the dynamic range of protein abundance, the efficiency of peptide digestion and ionization, and the stochastic nature of data-dependent acquisition. Second, and more importantly, our current exploration of the encystment mechanism is primarily based on correlative data derived from transcriptomic and proteomic analyses. While these multi-omics approaches provide powerful, system-wide insights and have allowed us to identify key candidate pathways (e.g., proteasome activity and specific metabolic shifts), the functional roles of these identified genes and proteins in driving the encystment process remain to be experimentally confirmed.

Future research should therefore be directed towards functional validation. Key candidate genes, particularly those encoding upregulated proteasome subunits or enzymes in the downregulated amino acid metabolism pathways, should be genetically manipulated (e.g., via gene knockout or RNAi) to assess their direct impact on encystment efficiency and cyst morphology. Furthermore, employing live-cell imaging or specific biochemical assays will be crucial to dynamically monitor these processes and establish causative, rather than correlative, relationships. Such experimental work is essential to move from observation to mechanistic understanding and to build a robust model of encystment regulation.

## ACKNOWLEDGMENTS

This work was supported by the National Natural Science Foundation of China (32570525, 32170446, and 32270512) and Distinguished Young Scholar and Interdisciplinary Cultivation Programs of Shandong University, China.

Designed research: Xinpeng Fan, Xiao Chen, Miaomiao Wang, and Juan Yang. Sample collection: Tian Wang and Zina Lin. Molecular experiment: Tao Hu and Tian Wang. Morphological observation: Tao Hu and Zina Lin. Bioinformatic analyses: Juan Yang. Visualization: Juan Yang and Miaomiao Wang. Statistical analyses: Zijia Liu, Juan Yang, and Miaomiao Wang. Supervision: Juan Yang, Miaomiao Wang, Xiao Chen, and Xinpeng Fan. Writing–original draft: Juan Yang and Miaomiao Wang. Writing–reviewing and editing: Juan Yang, Miaomiao Wang, Xiao Chen, and Xinpeng Fan. Funding acquisition: Xinpeng Fan and Xiao Chen.

The authors declare that they have no known competing financial interests or personal relationships that could have appeared to influence the work reported in this paper.

## AUTHOR AFFILIATIONS

[1]East China Normal University, School of Life Sciences, Shanghai, China
[2]Shandong University, Marine College, Weihai, China
[3]Shandong University, SDU-ANU Joint Science College, Weihai, China

## AUTHOR ORCIDs

Miaomiao Wang  http://orcid.org/0009-0003-6459-7352
Juan Yang  http://orcid.org/0000-0002-3478-606X
Xiao Chen  http://orcid.org/0000-0001-5059-8846
Xinpeng Fan  http://orcid.org/0000-0002-1822-7997

## FUNDING

| Funder | Grant(s) | Author(s) |
| --- | --- | --- |
| National Natural Science Foundation of China | 32570525, 32170446, 32270512 | Xinpeng Fan |

## AUTHOR CONTRIBUTIONS

Miaomiao Wang, Formal analysis, Methodology, Software, Visualization, Writing – original draft, Writing – review and editing | Tao Hu, Data curation, Methodology, Validation | Zina Lin, Data curation, Formal analysis, Methodology | Tian Wang, Data curation, Formal analysis | ZiJia Liu, Methodology | Xiao Chen, Conceptualization, Funding acquisition, Project administration, Supervision, Writing – review and editing.

## DATA AVAILABILITY

The Illumina sequencing data of the MAC genome have been deposited in the European Nucleotide Archive (ENA) under the project accession PRJEB108089, with the genome assembly accession number ERZ28945402, and additional assembly and annotation files are publicly available in Figshare at DOI:10.6084/m9.figshare.31415543. The RNA-seq reads have been deposited in GenBank under the accession number PRJNA1302648, and the proteomic data have been deposited in ProteomeXchange with the identifier PXD068223. Further detailed information is provided in Table S8.

## ADDITIONAL FILES

The following material is available online.

## Supplemental Material

**Supplemental Material (mSystems01757-25-s0001.docx).** Fig. S1 to S3; Tables S1 to S3.

**Table S4 (mSystems01757-25-s0002.xlsb).** Summary statistics of OrthoFinder orthogroup clustering for 32 species.

**Table S5 (mSystems01757-25-s0003.xlsb).** Integrated transcript expression, protein abundance, and functional annotation.

**Table S6 (mSystems01757-25-s0004.xlsx).** GO and KEGG pathway enrichment analysis results of DEGs during cystment in *O. granulifera*.

**Table S7 (mSystems01757-25-s0005.xlsx).** GO and KEGG pathway enrichment analysis results of differentially expressed proteins during cystment in *O. granulifera*.

**Table S8 (mSystems01757-25-s0006.xlsx).** Summary of accession numbers and data availability for genome, transcriptome, and proteome data sets in this study.

## Open Peer Review

**PEER REVIEW HISTORY (review-history.pdf).** An accounting of the reviewer comments and feedback.

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
