## [Reviewer comments · mSystems]

Probing the genomic and proteomic basis of encystment in *Oxytricha granulifera*

Miaomiao Wang, Juan Yang, Tao Hu, Zina Lin, Tian Wang, ZiJia Liu, Xiao Chen, and Xinpeng Fan

Corresponding Author(s): Xinpeng Fan, East China Normal University

Review Timeline:

Submission Date:	December 14, 2025
Editorial Decision:	February 5, 2026
Revision Received:	March 18, 2026
Accepted:	March 25, 2026

Editor: Germán Bonilla-Rosso

Reviewer(s): Disclosure of reviewer identity is with reference to reviewer comments included in decision letter(s). The following individuals involved in review of your submission have agreed to reveal their identity: Vojtech Zarsky (Reviewer #1)

Transaction Report:

DOI: <https://doi.org/10.1128/msystems.01757-25>

Re: mSystems01757-25 (**Probing the genomic and proteomic basis of encystment in *Oxytricha granulifera***)

Dear Prof. Xinpeng Fan:

Thank you for the privilege of reviewing your work. Below you will find my comments, instructions from the mSystems editorial office, and the reviewer comments. I extend an apology for the delay in the review, it has been difficult to secure reviewers with expertise in the genomics of an organism of such complexity, particularly over the winter holidays.

The responses of the reviewers are overall positive about this new version of the manuscript and agree on the relevance of your results. However, the reviewers still raise concerns regarding transparency, replicability and clarity.

Data availability. The final assembly is deposited in CNSA. I am unfamiliar with the repository, and it is not in the list of Data Repositories of ASM (<https://journals.asm.org/list-data-repositories>). Moreover, the website requires users to register and provide personal data, and a good fraction of the data deposited is not available or "controlled". And the reviewer link provided by the authors does not allow access to the data (including downloading) nor contains information about the availability upon release.

ASM's Open Data Policy (<https://journals.asm.org/open-data-policy>) establishes that a condition for publication is that authors make raw and processed data fully available and without restrictions. Data must be fully available during peer review upon request. Currently the best way to ensure your sequence data is shared and reused, is depositing in a database of the INSDC, namely GenBank, DDBK or ENA (<https://www.insdc.org/>). I would recommend the authors to deposit raw sequencing files in SRA, and expression data in GEO or ArrayExpress.

In addition the metadata, and particularly a table that cross-links gene identifiers and coordinates with contigs their annotations, is essential for reproducibility. For this multi-omics dataset, the table must also link across the datasets of genome assembly, transcriptomics and proteomics data. This is the only way to truly compare these different datasets.

The reviewers have also commented on the readability of the manuscript. Some of it comes from spelling and syntax mistakes (e.g. lines L234, L438, L636), or from a lack of clarity in the sentences (e.g. L237, L330, L348). I think some of the confusion also comes from the fact that there is a lot of referenced data from previous studies in the Results instead of the Introduction, and that these are being discussed in the Results instead of the Discussion (e.g. L336-338, L340-342, L370-376). An improvement in the compartmentalization of the manuscript will increase its readability and make it easier for both the reviewers and you intended readers.

Finally, please pay special attention to the description of the methodology as it is not entirely transparent, and it complicates the assessment of some claims formulated in the text that do not seem to align with the evidence presented. Below a few examples.

L347: Please explain how these results demonstrate the high quality of the genome. Particularly given the fact that you have no other reference genome to compare it and you have only 86.5% completeness.

L356: Was this really proteomes or translated protein-coding genes from genome assemblies? please clarify and correct if necessary.

L361-362: A high number of "species-specific" genes is expected when you are comparing very distant taxa.

Revision Guidelines

Sincerely,
Germán Bonilla-Rosso
Editor
mSystems

Reviewer #1 (Comments for the Author):

The re-submitted manuscript addresses mostly very well my previous notes and suggestions. I also appreciate that the changes were delivered in a very comprehensive way, including a file with marked changes. I believe that the general organization of the manuscript improved significantly.

There is, however, still one major issue that needs to be addressed: The predicted genes and their annotations must be provided! These are pivotal for all the analyses, and basically all the results (transcriptomics/proteomics expression analyses, orthologous group comparisons) don't make much sense without them. To be specific, both in the main text and in the tables, gene IDs (e.g. "gene18150") are used, however I don't see a way how to connect these IDs to real sequences and annotations. My apologies if I've missed these files in the repositories, but these should be certainly declared in the "Data availability" section. The provided reviewer link indeed brings me to the repository with the assembled contigs, however I don't see a way to access/download the file and I don't see any other files related to gene prediction and annotation.

Per-line (based on mSystems01757-25-Manuscript_Text_File.pdf):

- In 118: "...functional annotation of differentially expressed transcripts and proteins"
- In 178: The transcriptome assembly is already described in the previous section.
- In 327: "... de-duplicated ..."
- In 340: I believe EuGene (not eggNOG) was used to predict genes.
- In 348: "Ciliates often employ ..."
- In 352: "... Oxytricha ..."
- In 353-354: I suppose that since you identified proteins from MS data without major issues, this provides a support for the codon re-assignments.
- In 439: "... events during"
- In 486-490: This is a very long sentence and I have to admit that I have difficulties understanding it (I am not a native English speaker). Please consider re-phrasing the sentence and check the grammar.
- In 504: "... Pseudocohnilembus persalinus ..." (I believe it is appropriate to use the whole name here.)
- In 646: While Hsp70 is important for oxidative stress response, it isn't technically an oxidoreductase.
- In 646: If the GPX abbreviation is used for the first time, a full description should be also provided.
- In 704-711: I thank the authors for the review links, but they should be omitted in the final text.
- In 707: "The proteomics data ..."

Reviewer #2 (Comments for the Author):

I think the authors have sufficiently addressed my previous concerns.

I apologize to the authors for the delay in my response. It wasn't for the lack of planning on my part. I also applaud authors in taking on a study of encystment. It is a shame that so few ciliatologists are studying this phenomenon at the molecular level.

Reviewer #3 (Comments for the Author):

This manuscript presents an ambitious and potentially valuable multi-omics study of encystment in *Oxytricha granulifera*, integrating genome assembly, transcriptomic, proteomic, and morphological analyses. The dataset is substantial, and the biological question is well aligned with the scope of mSystems. That said, despite improvements relative to earlier versions, the manuscript would benefit from further revision to improve methodological transparency, statistical clarity, and consistency in the description of analytical pipelines, particularly for RNA-seq differential expression, alternative splicing, and proteomics. In addition, several conclusions rely on correlative evidence and should be more carefully framed to avoid over-interpretation. Addressing these points would substantially strengthen the rigor, reproducibility, and overall impact of the study.

General Comment for the Authors

This manuscript presents an ambitious and potentially valuable multi-omics study of encystment in *Oxytricha granulifera*, integrating genome assembly, transcriptomic, proteomic, and morphological analyses. The dataset is substantial, and the biological question is well aligned with the scope of *mSystems*. That said, despite improvements relative to earlier versions, the manuscript would benefit from further revision to improve methodological transparency, statistical clarity, and consistency in the description of analytical pipelines, particularly for RNA-seq differential expression, alternative splicing, and proteomics. In addition, several conclusions rely on correlative evidence and should be more carefully framed to avoid over-interpretation. Addressing these points would substantially strengthen the rigor, reproducibility, and overall impact of the study.

Overall Assessment

Strengths

- The study addresses an important gap in protist biology and dormancy research.
- Generation of a new *Oxytricha* genome and integration with transcriptomic and proteomic data is valuable.
- Morphological observations are well documented and support the general biological narrative.

Major Weaknesses

- Key methods remain insufficiently described for reproducibility (RNA-seq statistics, alternative splicing, proteomics).
- A critical statistical inconsistency remains in the differential expression analysis description.
- Encystment induction and cyst validation lack quantitative rigor.
- Language and structure often emphasize data generation rather than biological insight.

Major Issues (must be addressed)

1. Incorrect and inconsistent statistical methodology for RNA-seq

Lines 212–214: The description of differential expression analysis is unclear. The use of Student's *t*-test is stated, followed by DESeq2, which does not rely on *t*-tests. This is not a minor wording issue but a methodological problem that affects confidence in the differential expression analysis. The authors must clearly describe the statistical model actually used, remove incorrect statements, and clarify all filtering and normalization steps.

2. Insufficient definition of biological replication

The manuscript repeatedly refers to “three replicates” for transcriptomic, proteomic, and qPCR analyses without clearly specifying whether these represent independent biological replicates (separate cultures or encystment inductions) or technical replicates. In the qPCR section, replication appears to refer only to technical wells. This ambiguity affects the interpretation of all downstream analyses and should be clarified explicitly.

3. Encystment induction and cyst validation lack quantitative support

Lines 131–137: Encystment is induced by prolonged starvation and cysts are subsequently used for transcriptomic and proteomic analyses; however, no quantitative assessment of encystment

efficiency, cyst maturity, or dormancy state is provided. Although morphological observations (light microscopy and electron microscopy) are consistent with encystment, these data are qualitative and do not allow assessment of the homogeneity and biological comparability of the cyst samples used for omics analyses. To strengthen this aspect, the authors could (i) report the proportion of cells exhibiting cyst morphology after 30 days of starvation, (ii) include a simple excystment assay to confirm cyst viability and dormancy (strongly encouraged if feasible), or (iii) explicitly acknowledge this limitation if additional validation is not available.

4. Centrifugation and solution reporting

Centrifugation conditions are reported in rpm without specifying the rotor type or radius. Because rpm does not uniquely define the applied centrifugal force, this information is insufficient for reproducibility. The authors should report centrifugation conditions in relative centrifugal force ($\times g$) and specify the rotor type (fixed-angle or swinging-bucket) used. In addition, where solutions or reagents are prepared, reporting final concentrations rather than procedural volumes would improve clarity and reproducibility.

5. Alternative splicing analysis

Lines 227–237: Although an “Analysis of alternative splicing” section has been added, the description remains insufficient for reproducibility. While the use of StringTie and gffcompare is specified, the authors do not indicate which R packages or scripts were used, the statistical criteria applied, read coverage or expression thresholds, or how alternative splicing events were quantitatively compared between conditions. Simply referring to “established bioinformatics pipelines in R” is insufficient to reproduce the analysis.

6. AMT / 6mA over-interpretation

The manuscript states that the methyltransferases responsible for 6mA methylation in *O. granulifera* were identified; however, the evidence provided is limited to domain annotation, phylogenetic inference, and differential gene expression of AMT family homologs. No direct measurement of 6mA levels, functional perturbation of AMT genes, or biochemical validation of methyltransferase activity is presented. The data therefore support identification of putative AMT family homologs with conserved domains and expression patterns consistent with a potential role in 6mA regulation, but do not demonstrate enzymatic responsibility or causality. The wording and conclusions should be revised to avoid over-interpretation.

7. KEGG interpretation and cross-omics ambiguity / proteomics methodological transparency

Results ~330–520; Methods 204–207, 224–225, 252–272: KEGG pathway enrichment analyses are performed separately for transcriptomic and proteomic datasets (Supplementary Figures S2 and S3, respectively); however, KEGG pathways are frequently discussed without clearly distinguishing whether the supporting evidence derives from RNA-seq data, proteomics data, or both. In addition, the methodological description of KEGG enrichment is insufficient for both datasets, including the KEGG database version, enrichment strategy, statistical testing framework, background set definition, and multiple-testing correction. These interpretive ambiguities are compounded by limited methodological transparency in the proteomics analysis, which does not report the protein sequence database used for searching, false discovery rate (FDR) thresholds at peptide and protein levels, normalization procedures, or the statistical criteria used to define differential protein abundance. Together, this limits reproducibility and weakens the strength of the claimed cross-omics pathway concordance.

8. Genome assembly description lacks key context for ciliates

Lines 178–180: Although the Introduction states that a macronuclear genome was assembled, the Methods do not explicitly describe how macronuclear DNA was obtained or whether micronuclear DNA was excluded. Given the nuclear dualism of ciliates, it should be clearly stated whether the assembled genome represents macronuclear DNA exclusively and how this was ensured. In addition, basic sequencing metrics such as sequencing depth and estimated genome coverage are not explicitly reported, limiting assessment of genome assembly completeness and reliability.

Minor Issues and Clarifications

Language and clarity

The manuscript contains frequent grammatical errors, awkward phrasing, and redundant sentences. Several sentences are overly long and obscure the main message. In addition, specialized terminology and acronyms are often used without definition at first mention, which may limit accessibility for a general microbiology audience. Terms such as “4D label-free proteomics” can be briefly explained. All acronyms and technical terms should be defined upon first use.

Microscopy

While multiple microscopy techniques are used to illustrate morphological changes during encystment, the manuscript does not report the number of cells or sections examined for each technique, nor does it clarify how images were selected. It is therefore unclear whether the presented images are representative observations or whether any quantitative assessment was performed. In addition, the imaging conditions for Figure 1A and 1B are not explicitly stated in the figure legend, making it difficult to determine that these panels represent unstained bright-field images and do not involve any staining. Reporting the number of cells or sections analyzed, clarifying image selection criteria, and providing more explicit figure legends would improve the rigor and interpretability of the microscopy data.

RNA-seq library preparation

Lines 164–171: The description of RNA-seq library preparation is repetitive and unclear, with reverse transcription and PCR amplification steps described twice in succession. It is also not stated whether the libraries were strand-specific, which is important for interpreting transcript structure and alternative splicing analyses. Finally, basic sequencing metrics, including sequencing depth and read counts per sample, are not reported and should be included to allow proper assessment of data quality and comparability across samples.

Data availability

Lines 706–708: The manuscript states that “transcriptomics data” were deposited in ProteomeXchange (PXD068223); however, ProteomeXchange/PRIDE hosts proteomics datasets. This appears to be a wording error and should be corrected for clarity.

Recommendation:

The manuscript would benefit from further revision to improve methodological transparency, statistical description, and clarity of analysis. In particular, clearer reporting of RNA-seq and proteomics workflows, replication, and statistical methods is needed, and inclusion of a more quantitative assessment of cyst sample homogeneity (as outlined above) would strengthen the rigor of the study.

Dr Germán Bonilla-Rosso
Editor, mSystems
14-Mar-2026

R: Probing the genomic and proteomic basis of encystment in *Oxytricha granulifera* Manuscript
Number: mSystems01757-25

Dear Dr. Bonilla-Rosso

We sincerely appreciate the journal's interest in our manuscript, and you and the reviewers for the time and effort spent on its evaluation, as well as the constructive feedback provided.

We have carefully considered all the comments and have undertaken extensive revisions to address the concerns raised, particularly regarding the availability of data and methods to ensure reproducibility. A detailed description has been included in the revised manuscript to clearly indicate the location where all relevant data can be accessed. We believe these revisions have substantially strengthened the readability of manuscript and fully addressed the issues raised by the reviewers and yourself. We are confident that the revised version now meets the high standards of mSystems and would be of significant interest to your readers.

Thank you once again for the opportunity to resubmit our work. We look forward to hearing from you.

Sincerely,
Xinpeng FAN

Response to editor and reviewers

Editor

The responses of the reviewers are overall positive about this new version of the manuscript and agree on the relevance of your results. However, the reviewers still raise concerns regarding transparency, replicability and clarity.

Data availability. The final assembly is deposited in CNSA. I am unfamiliar with the repository, and it is not in the list of Data Repositories of ASM (<https://journals.asm.org/list-data-repositories>). Moreover, the website requires users to register and provide personal data, and a good fraction of the data deposited is not available or "controlled". And the reviewer link provided by the authors does not allow access to the data (including downloading) nor contains information about the availability upon release.

ASM's Open Data Policy (<https://journals.asm.org/open-data-policy>) establishes that a condition for publication is that authors make raw and processed data fully available and without restrictions. Data must be fully available during peer review upon request. Currently the best way to ensure your sequence data is shared and reused, is depositing in a database of the INSDC, namely GenBank, DDBK or ENA (<https://www.insdc.org/>). I would recommend the authors to deposit raw sequencing files in SRA, and expression data in GEO or ArrayExpress.

In addition the metadata, and particularly a table that cross-links gene identifiers and coordinates

with contigs their annotations, is essential for reproducibility. For this multi-omics dataset, the table must also link across the datasets of genome assembly, transcriptomics and proteomics data. This is the only way to truly compare these different datasets.

R: Thank you for this important suggestion. We have re-uploaded the updated genome data to the ENA database under the project accession PRJEB108089.

To ensure reproducibility and enable cross-dataset comparison, we provide Table S8 that clearly lists the detailed repository locations of all multi-omics datasets, including genome assembly, transcriptomics, and proteomics data.

In addition, Table S5 systematically links transcriptomic and proteomic analysis results to functional annotations.

These tables allow direct and reliable comparison of the different omics layers as recommended.

The reviewers have also commented on the readability of the manuscript. Some of it comes from spelling and syntax mistakes (e.g. lines L234, L438, L636), or from a lack of clarity in the sentences (e.g L237, L330, L348). I think some of the confusion also comes from the fact that there is a lot of referenced data from previous studies in the Results instead of the Introduction, and that these are being discussed in the Results instead of the Discussion (e.g. L336-338, L340-342, L370-376). An improvement in the compartmentalization of the manuscript will increase its readability and make it easier for both the reviewers and you intended readers.

R: We thank the reviewer for these valuable suggestions to improve the manuscript's readability and structure. We have carefully revised the manuscript accordingly:

We have corrected the specific errors at lines L234, L438, and L636, and have thoroughly proofread the entire manuscript for additional errors. The sentences at lines L237, L330, and L348 have been rewritten for clarity. And we agree that the Results section contains excessive references and discussion content. Relocated discussion points (e.g., L370-376) to the Discussion section. After re-evaluating lines L336-338, we agreed that their relevance to the main findings was limited. We have therefore removed these lines to improve the focus and readability of the manuscript, and the lines L340-342 has been adjusted accordingly. These revisions have significantly improved the logical flow and readability of the manuscript. All changes are marked in the revised text.

Finally, please pay special attention to the description of the methodology as it is not entirely transparent, and it complicates the assessment of some claims formulated in the text that do not seem to align with the evidence presented. Below a few examples.

R: We sincerely appreciate the editor's careful and constructive comment. We have carefully revised and improved the transparency of all methodology descriptions throughout the manuscript, ensuring that each step is clearly and explicitly stated. Correspondingly, we have also carefully checked and adjusted the relevant statements in the text to ensure they are fully consistent and well supported by the presented results and evidence. All revisions have been made to enhance the rigor, clarity, and reproducibility of the study.

L347: Please explain how these results demonstrate the high quality of the genome. Particularly given the fact that you have no other reference genome to compare it and you have only 86.5% completeness.

R: We agree with the reviewer. This statement has been removed from the revised manuscript.

L356: Was this really proteomes or translated protein-coding genes from genome assemblies? please clarify and correct if necessary.

R: We thank the reviewer for this careful clarification. The analysis of codon usage was based on predicted protein-coding genes from our genome assembly, not on experimental proteomics data. While our study includes proteomic data for differential expression analysis, definitive confirmation of codon reassignment would require targeted mass spectrometry analysis to identify specific read-through peptides at stop codon positions, which was beyond the scope of this study. We have corrected the terminology in the revised manuscript to avoid ambiguity.

L361-362: A high number of "species-specific" genes is expected when you are comparing very distant taxa.

R: We thank the reviewer for this important comment. We fully agree that a high number of species-specific genes is common and expected in gene family expansion and contraction analysis when comparing distantly related taxa, because the large evolutionary divergence makes it difficult to identify homologous genes across distant species, rather than reflecting genuine massive novel gene origination. We have added a clarification in the revised manuscript to avoid misinterpretation.

Reviewer #1

The re-submitted manuscript addresses mostly very well my previous notes and suggestions. I also appreciate that the changes were delivered in a very comprehensive way, including a file with marked changes. I believe that the general organization of the manuscript improved significantly. There is, however, still one major issue that needs to be addressed: The predicted genes and their annotations must be provided! These are pivotal for all the analyses, and basically all the results (transcriptomics/proteomics expression analyses, orthologous group comparisons) don't make much sense without them. To be specific, both in the main text and in the tables, gene IDs (e.g. "gene18150") are used, however I don't see a way how to connect these IDs to real sequences and annotations. My apologies if I've missed these files in the repositories, but these should be certainly declared in the "Data availability" section. The provided reviewer link indeed brings me to the repository with the assembled contigs, however I don't see a way to access/download the file and I don't see any other files related to gene prediction and annotation.

R: Thank you for this important comment. We have provided Table S9 that links each gene identifier to its corresponding protein and functional annotation for easy lookup. Due to file format constraints, the genome annotation files are publicly available at FigShare:

[10.6084/m9.figshare.31415543](https://figshare.com/10.6084/m9.figshare.31415543)

A reviewer link is also provided: <https://figshare.com/s/5e01e7ae2d8b6b33c16d>.

Per-line (based on mSystems01757-25-Manuscript_Text_File.pdf):

- In 118: "...functional annotation of differentially expressed transcripts and proteins"

R: We have revised it as suggested.

- In 178: The transcriptome assembly is already described in the previous section.

R: We thank the reviewer for catching this error. We have revised the RNA-seq methods section by removing redundant sentences to improve clarity.

- In 327: "... de-duplicated ..."

R: We have revised it as suggested.

- In 340: I believe EuGene (not eggNOG) was used to predict genes.

R: We thank the reviewer for catching this error. It was corrected as suggested.

- In 348: "Ciliates often employ ..."

R: We have revised it as suggested.

- In 352: "... Oxytricha ..."

R: We thank the reviewer for catching this error. We have revised it as suggested.

- In 353-354: I suppose that since you identified proteins from MS data without major issues, this provides a support for the codon re-assignments.

Re: Thank you for pointing this out. Mass spectrometry identifies peptides, not full proteins. A successful match of partial peptides is sufficient to assign them to a gene, but it only confirms the existence of those specific peptide sequences—it does not provide information about the encoding DNA sequence. Therefore, while the MS data support the expression of these genomic regions, they cannot be used to infer or validate the codon assignments. The inferred stop codon reassignments are strictly derived from genome assembly and gene structure prediction, independent of the MS results.

- In 439: "... events during"

R: We have revised it as suggested.

- In 486-490: This is a very long sentence and I have to admit that I have difficulties understanding it (I am not a native English speaker). Please consider re-phrasing the sentence and check the grammar.

R: We sincerely apologize for the unclear and overly lengthy sentence that caused difficulty in understanding. We have carefully rephrased the sentence to improve its clarity, readability, and grammatical accuracy in the revised manuscript.

- In 504: "... Pseudocohnilembus persalinus ..." (I believe it is appropriate to use the whole name here.)

R: We have revised it as suggested.

- In 646: While Hsp70 is important for oxidative stress response, it isn't technically an oxidoreductase.

R: We thank the reviewer for this careful correction. Hsp70 is indeed a molecular chaperone rather than an oxidoreductase. We have revised the description accordingly to avoid misclassification.

- In 646: If the GPX abbreviation is used for the first time, a full description should be also provided.

R: We apologize for the missing full name of the abbreviation. The full name glutathione peroxidase (GPX) has been added at its first occurrence in the revised manuscript.

- In 704-711: I thank the authors for the reviewe links, but they should be omitted in the final text.

R: We agree with the reviewer's suggestion. These links will be removed in the final published version of the manuscript.

- In 707: "The proteomics data ..."

R: We thank the reviewer for catching this error. We have revised it as suggested.

Reviewer #2 (Comments for the Author):

I think the authors have sufficiently addressed my previous concerns.

I apologize to the authors for the delay in my response. It wasn't for the lack of planning on my part. I also applaud authors in taking on a study of encystment. It is a shame that so few ciliatologists are studying this phenomenon at the molecular level.

R: We sincerely appreciate the reviewer's thoughtful comments and careful evaluation of our work. We are grateful that our revisions have addressed the previous concerns satisfactorily. We also greatly value the reviewer's recognition of our study on encystment in ciliates, which we hope will contribute to the limited molecular understanding of this important biological process.

Reviewer #3 (Comments for the Author):

This manuscript presents an ambitious and potentially valuable multi-omics study of encystment in *Oxytricha granulifera*, integrating genome assembly, transcriptomic, proteomic, and morphological analyses. The dataset is substantial, and the biological question is well aligned with the scope of mSystems. That said, despite improvements relative to earlier versions, the manuscript would benefit from further revision to improve methodological transparency, statistical clarity, and consistency in the description of analytical pipelines, particularly for RNA-seq differential expression, alternative splicing, and proteomics. In addition, several conclusions rely on correlative evidence and should be more carefully framed to avoid over-interpretation. Addressing these points would substantially strengthen the rigor, reproducibility, and overall impact of the study.

R: We greatly appreciate the reviewer's careful and constructive evaluation of our manuscript. We are grateful for the positive comments on the multi-omics dataset and the scientific significance of our study on encystment in *Oxytricha granulifera*. According to the reviewer's suggestions, we have further revised the manuscript to enhance the transparency and clarity of all methodologies, especially for the statistical analysis and analytical pipelines of RNA-seq differential expression, alternative splicing, and proteomics. We have also carefully rephrased several conclusions to

avoid over-interpretation and ensure they are appropriately framed based on the correlative evidence provided. These revisions have greatly improved the rigor, reproducibility, and scientific quality of the work.

Major Issues (must be addressed)

1. Incorrect and inconsistent statistical methodology for RNA-seq

Lines 212–214: The description of differential expression analysis is unclear. The use of Student's t-test is stated, followed by DESeq2, which does not rely on t-tests. This is not a minor wording issue but a methodological problem that affects confidence in the differential expression analysis. The authors must clearly describe the statistical model actually used, remove incorrect statements, and clarify all filtering and normalization steps.

R: We appreciate the reviewer's careful comment on the statistical methodology for RNA-seq analysis. In our initial analysis, we performed differential gene identification using both Student's t-test and DESeq2, and initially presented results based on the Student's t-test. To ensure consistency and rigor in transcriptome analysis, we have now unified all differential expression results and descriptions using the DESeq2 pipeline throughout the manuscript. The detailed normalization, filtering, and statistical modeling procedures have been clearly described to ensure accuracy and reproducibility.

2. Insufficient definition of biological replication

The manuscript repeatedly refers to "three replicates" for transcriptomic, proteomic, and qPCR analyses without clearly specifying whether these represent independent biological replicates (separate cultures or encystment inductions) or technical replicates. In the qPCR section, replication appears to refer only to technical wells. This ambiguity affects the interpretation of all downstream analyses and should be clarified explicitly.

R: We thank the reviewer for highlighting this ambiguity. We have now clarified the replication strategy in the revised manuscript:

Transcriptomics and proteomics: The "three replicates" refer to independent biological replicates, each derived from separate cultures and independent encystment inductions. For each replicate, approximately 1.5×10^6 cells were collected.

qPCR: The "three replicates" in the qPCR analysis refer to technical replicates (three parallel wells from the same biological sample).

3. Encystment induction and cyst validation lack quantitative support

Lines 131–137: Encystment is induced by prolonged starvation and cysts are subsequently used for transcriptomic and proteomic analyses; however, no quantitative assessment of encystment efficiency, cyst maturity, or dormancy state is provided. Although morphological observations (light microscopy and electron microscopy) are consistent with encystment, these data are qualitative and do not allow assessment of the homogeneity and biological comparability of the cyst samples used for omics analyses. To strengthen this aspect, the authors could (i) report the proportion of cells exhibiting cyst morphology after 30 days of starvation, (ii) include a simple encystment assay to confirm cyst viability and dormancy (strongly encouraged if feasible), or (iii) explicitly acknowledge this limitation if additional validation is not available.

R: We thank the reviewer for this constructive suggestion. Following the reviewer's advice, we

have now included quantitative data on encystment efficiency and performed an excystment assay to confirm cyst viability and dormancy.

Encystment efficiency and homogeneity: Microscopic examination after 30 days of starvation revealed that 100% of cells had differentiated into mature cysts, confirming the homogeneity of the cyst samples used for omics analyses.

Cyst viability and dormancy (excystment assay): we performed excystment assay by transferring mature cysts to fresh culture medium. Approximately 85% (17–18 out of 20 cysts) successfully excysted and resumed vegetative growth, confirming that the cysts were viable and had entered a true dormant state prior to omics analysis.

4. Centrifugation and solution reporting

Centrifugation conditions are reported in rpm without specifying the rotor type or radius. Because rpm does not uniquely define the applied centrifugal force, this information is insufficient for reproducibility. The authors should report centrifugation conditions in relative centrifugal force ($\times g$) and specify the rotor type (fixed-angle or swinging-bucket) used. In addition, where solutions or reagents are prepared, reporting final concentrations rather than procedural volumes would improve clarity and reproducibility.

R: We thank the reviewer for this suggestion. According to the reviewer's valuable comments, we have revised the manuscript to improve the experimental reproducibility and clarity. All centrifugation conditions have been converted from rpm to relative centrifugal force ($\times g$) with the corresponding rotor information specified. In addition, we have updated the description of qRT-PCR reagents to report final concentrations instead of only procedural volumes. All revisions have been carefully checked and highlighted in the revised manuscript.

5. Alternative splicing analysis

Lines 227–237: Although an “Analysis of alternative splicing” section has been added, the description remains insufficient for reproducibility. While the use of StringTie and gffcompare is specified, the authors do not indicate which R packages or scripts were used, the statistical criteria applied, read coverage or expression thresholds, or how alternative splicing events were quantitatively compared between conditions. Simply referring to “established bioinformatics pipelines in R” is insufficient to reproduce the analysis.

R: Thank you for the valuable suggestion. We have now added more detailed information in the corresponding section of the manuscript to improve the reproducibility of our alternative splicing analysis.

6. AMT / 6mA over-interpretation

The manuscript states that the methyltransferases responsible for 6mA methylation in *O. granulifera* were identified; however, the evidence provided is limited to domain annotation, phylogenetic inference, and differential gene expression of AMT family homologs. No direct measurement of 6mA levels, functional perturbation of AMT genes, or biochemical validation of methyltransferase activity is presented. The data therefore support identification of putative AMT family homologs with conserved domains and expression patterns consistent with a potential role in 6mA regulation, but do not demonstrate enzymatic responsibility or causality. The wording and conclusions should be revised to avoid over-interpretation.

R: We thank the reviewer for this critical and helpful comment. We agree that our initial wording overinterpreted the data. We have revised the manuscript to more accurately reflect that our findings identify putative AMT family homologs with conserved domains and expression patterns consistent with a potential role in 6mA regulation, rather than demonstrating direct enzymatic responsibility.

7. KEGG interpretation and cross-omics ambiguity / proteomics methodological transparency

Results ~330–520; Methods 204–207, 224–225, 252–272: KEGG pathway enrichment analyses are performed separately for transcriptomic and proteomic datasets (Supplementary Figures S2 and S3, respectively); however, KEGG pathways are frequently discussed without clearly distinguishing whether the supporting evidence derives from RNA-seq data, proteomics data, or both. In addition, the methodological description of KEGG enrichment is insufficient for both datasets, including the KEGG database version, enrichment strategy, statistical testing framework, background set definition, and multiple-testing correction. These interpretive ambiguities are compounded by limited methodological transparency in the proteomics analysis, which does not report the protein sequence database used for searching, false discovery rate (FDR) thresholds at peptide and protein levels, normalization procedures, or the statistical criteria used to define differential protein abundance. Together, this limits reproducibility and weakens the strength of the claimed cross-omics pathway concordance.

R: Thank you for the valuable comment. We have supplemented the detailed methodology of KEGG enrichment analysis in the revised manuscript. KEGG enrichment analysis was performed using TBtools based on EggNOG annotations. The background gene set was defined as all protein-coding genes of the protist species annotated with KEGG pathways (<https://www.genome.jp/kegg/tables/br08606.html#4>). A threshold of $P < 0.05$ was applied to identify significantly enriched pathways. In addition, we have carefully revised the text to clearly distinguish whether KEGG pathway results are supported by transcriptomic data, proteomic data, or both throughout the Results and Discussion sections.

8. Genome assembly description lacks key context for ciliates

Lines 178–180: Although the Introduction states that a macronuclear genome was assembled, the Methods do not explicitly describe how macronuclear DNA was obtained or whether micronuclear DNA was excluded. Given the nuclear dualism of ciliates, it should be clearly stated whether the assembled genome represents macronuclear DNA exclusively and how this was ensured. In addition, basic sequencing metrics such as sequencing depth and estimated genome coverage are not explicitly reported, limiting assessment of genome assembly completeness and reliability.

R: We thank the reviewer for raising this important point regarding nuclear dualism in ciliates. In our study, total genomic DNA was extracted from vegetative cells without physical separation of macronuclei and micronuclei. However, based on the well-established biological features of ciliates, we are confident that the assembled genome predominantly represents the macronuclear genome for the following reasons:

(i) Polyploidy of the macronucleus: *Oxytricha* (e.g., *Oxytricha trifallax*) maintains its macronuclear chromosomes at an extremely high copy number, averaging ~2,000 copies per gene. In contrast, the micronucleus is diploid (2C). This ~1000-fold difference in stoichiometry ensures that the sequencing library is overwhelmingly dominated by macronuclear DNA.

(ii) Extensive DNA elimination during macronuclear development: In spirotrichous ciliates such as *Euplotes*, up to 80-95% of the micronuclear genome (including internally eliminated sequences, IESs, and transposons) is eliminated during macronuclear development. The resulting macronuclear genome is highly streamlined and gene-dense. Even if micronuclear sequences were present in the DNA extract, their representation in the sequencing library would be minimal due to their low copy number and high repeat content.

(iii) Post-assembly validation: *Oxytricha* macronuclear chromosomes are typically gene-sized "nanochromosomes" capped with telomeres. Examination of scaffold ends revealed the presence of telomeric repeats (CCCCAAA), a hallmark of macronuclear chromosomes in ciliates.

(iv) Sequencing coverage: We generated 15 Gb of raw sequencing data. Given the assembled genome size of 41.7 Mb, the estimated sequencing depth is approximately 360×. Such high coverage is consistent with the amplification levels of the macronucleus, whereas any residual micronuclear DNA would be present at extremely low coverage (<1× relative to MAC) and thus excluded during the assembly process or filtered out as low-coverage noise. We have added the following description to the "Genomic DNA sequencing and de novo assembly" section.

Minor Issues and Clarifications

Language and clarity

The manuscript contains frequent grammatical errors, awkward phrasing, and redundant sentences. Several sentences are overly long and obscure the main message. In addition, specialized terminology and acronyms are often used without definition at first mention, which may limit accessibility for a general microbiology audience. Terms such as "4D label-free proteomics" can be briefly explained. All acronyms and technical terms should be defined upon first use.

R: We sincerely appreciate the editor's careful and constructive comments on the language clarity and terminology of our manuscript. We have carefully revised the entire manuscript to correct grammatical errors, eliminate awkward phrasing and redundant sentences, and simplify overly long sentences to improve readability and ensure the main points are clearly conveyed. In addition, we have defined all specialized terms and acronyms upon their first mention, these revisions have greatly improved the clarity, fluency, and overall quality of the manuscript.

Microscopy

While multiple microscopy techniques are used to illustrate morphological changes during encystment, the manuscript does not report the number of cells or sections examined for each technique, nor does it clarify how images were selected. It is therefore unclear whether the presented images are representative observations or whether any quantitative assessment was performed. In addition, the imaging conditions for Figure 1A and 1B are not explicitly stated in the figure legend, making it difficult to determine that these panels represent unstained bright-field images and do not involve any staining.

Reporting the number of cells or sections analyzed, clarifying image selection criteria, and providing more explicit figure legends would improve the rigor and interpretability of the microscopy data.

R: We thank the reviewer for this valuable suggestion. We have now clarified the microscopy data as follows:

Number of cells/sections analyzed: For light microscopy, >25 cells from three independent replicates were examined. For TEM, 20 sections of 5 individuals were analyzed. This information has been added to the Results section. All images shown are representative of the predominant morphology observed. In addition, the imaging conditions for Figure 1A and 1B have been explicitly described in the corresponding figure legend to improve the rigor and interpretability of the data.

RNA-seq library preparation

Lines 164–171: The description of RNA-seq library preparation is repetitive and unclear, with reverse transcription and PCR amplification steps described twice in succession. It is also not stated whether the libraries were strand-specific, which is important for interpreting transcript structure and alternative splicing analyses. Finally, basic sequencing metrics, including sequencing depth and read counts per sample, are not reported and should be included to allow proper assessment of data quality and comparability across samples.

R: We thank the reviewer for these helpful comments. We have revised the RNA-seq Methods section to eliminate redundancy and to clearly state that the libraries were constructed using a strand-specific protocol. Specifically, we used a stranded mRNA-seq workflow in which the second cDNA strand was selectively marked and removed (dUTP-based), thereby preserving transcript orientation information. This strengthens transcript structure interpretation and alternative splicing analyses. We also added basic sequencing metrics to the Methods: on average, approximately 6 Gb of clean paired-end data per sample (Q30 > 90%) were obtained across three biological replicates per condition, ensuring sufficient depth and comparability.

Data availability

Lines 706–708: The manuscript states that “transcriptomics data” were deposited in ProteomeXchange (PXD068223); however, ProteomeXchange/PRIDE hosts proteomics datasets. This appears to be a wording error and should be corrected for clarity

R: Thank you for catching this error. The wording has been corrected. The proteomics data were deposited in the ProteomeXchange Consortium via the PRIDE partner repository under accession number PXD068223.

Re: mSystems01757-25R1 (**Probing the genomic and proteomic basis of encystment in *Oxytricha granulifera***)

Dear Prof. Xinpeng Fan:

Your manuscript has been accepted, and I am forwarding it to the ASM production staff for publication. Your paper will first be checked to make sure all elements meet the technical requirements. ASM staff will contact you if anything needs to be revised before copyediting and production can begin. Otherwise, you will be notified when your proofs are ready to be viewed.

Cover Image Submissions: If you would like to submit a potential Cover Image, please email a file and a short legend to mSystems@asmusa.org. Please note that we can only consider images that (i) the authors created or own and (ii) have not been previously published. By submitting, you agree that the image can be used under the same terms as the published article. Image File requirements: TIF/EPS, 7.5 inches wide by 8.25 inches tall (at least 2,250 pixels wide by 2,475 pixels tall), minimum 300 dpi resolution (600 dpi preferred), RGB, and no figure elements, e.g., arrows or panel labels. The legend should be a short description of the image, 1-2 sentences recommended. Please download and use this interactive template in Adobe to ensure that your proposed cover image meets our size requirements (<https://journals.asm.org/pb-assets/pdf-text-excel-files/ASM-Interactive-Sizing-Cover-Template-1715689791.pdf>).

Sincerely,
Germán Bonilla-Rosso
Editor
mSystems